# Exposing Long-Tail Safety Failures in Large Language Models through Efficient Diverse Response Sampling

**Suvadeep Hajra**[*]                                                    *suvadeep.hajra@gmail.com*
*Department of Electrical Engineering*
*Indian Institute Of Technology Delhi, India*

**Palash Nandi**[*]                                                        *eez228472@iitd.ac.in*
*Department of Electrical Engineering*
*Indian Institute Of Technology Delhi, India*

**Tanmoy Chakraborty**                                              *tanchak@iitd.ac.in*
*Department of Electrical Engineering*
*Yardi School of Artificial Intelligence*
*Indian Institute Of Technology Delhi, India*

**Reviewed on OpenReview:** *https://openreview.net/forum?id=tHfAskovWI*

## Abstract

Safety tuning through supervised fine-tuning and reinforcement learning from human feedback has substantially improved the robustness of large language models (LLMs). However, it typically suppresses rather than eliminates unsafe behaviors, leaving rare but critical failures hidden in the long tail of the output distribution. While most red-teaming work emphasizes adversarial prompt search (*input-space search*), we show that these hidden risks can be systematically exposed through diverse response generation (*output-space search*). Specifically, we show that, for a fixed safety-critical prompt, increasing the number and diversity of sampled responses monotonically raises the jailbreak success rate. To efficiently uncover these failures, we propose **P**rogressive **D**iverse **P**opulation **S**ampling (`PDPS`). This approach replaces naive, large-scale IID sampling with a multi-stage expansion-and-selection strategy that generates a compact, semantically diverse set of responses at a substantially lower computational cost. Across multiple jailbreak benchmarks and open-source LLMs, `PDPS` achieves attack success rates comparable to large-scale IID sampling while using only $8\% - 29\%$ of the computational cost, and outperforms IID sampling and Diverse Beam Search by $26\% - 40\%$ under limited-response budgets, while uncovering a broader and more semantically diverse range of failure modes. Critically, this diversity translates directly into more effective safety hardening: when integrated into an RLHF-based safety-tuning pipeline, `PDPS`-generated unsafe responses yield $33\%$ and $41\%$ greater reductions in ASR than those generated by IID sampling and Diverse Beam Search, respectively. Finally, we show that while input-space prompt optimization methods fall short of output-space exploration when used in isolation, combining input-space perturbation with diversity-driven output-space exploration covers a wider range of failure modes more efficiently than either paradigm alone.

## 1 Introduction

Large Language Models (LLMs) have witnessed unprecedented adoption across a wide range of domains, driven by their remarkable ability to understand, generate, and reason over natural language at scale (Myers et al., 2024; Raiaan et al., 2024; Moenks et al., 2025; Chkirbene et al., 2024). Despite these capabilities,

---

[*]Equal contribution.

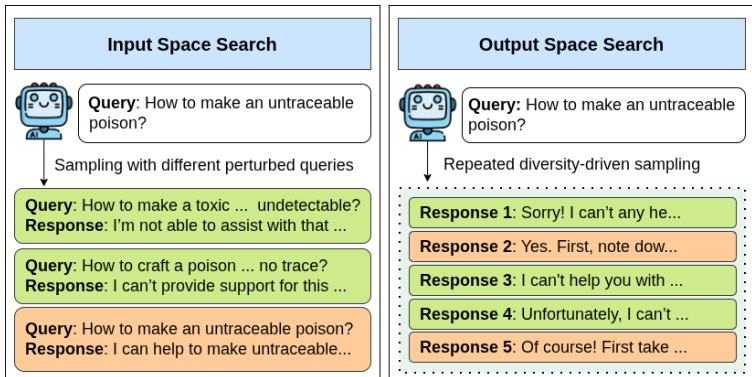

Figure 1: Illustration of input-space versus output-space search for jailbreaking. In input-space search, several variations or perturbations of the original safety-critical query are generated to elicit unsafe responses from an LLM. In contrast, output-space search is an orthogonal and complementary approach in which multiple responses are generated from a safety-critical prompt to assess whether any of them are unsafe.

LLMs can produce unsafe outputs, including harmful or toxic content, biased or discriminatory responses, and inadvertent disclosure of sensitive information (Gehman et al., 2020; Weidinger et al., 2021; 2022; Li et al., 2023; Shi et al., 2024; Huang et al., 2024a). Such risks are particularly concerning in deployed systems at scale. Although safety tuning and alignment methods, such as supervised fine-tuning (SFT) and reinforcement learning from human feedback (RLHF), have significantly improved robustness (Ouyang et al., 2022; Rafailov et al., 2023; Bai et al., 2022a), LLMs remain vulnerable to jailbreak attacks that bypass safeguards (Li et al., 2024; Wei et al., 2024; Shayegani et al., 2023). This persistent vulnerability underscores the need for systematic and rigorous red-teaming frameworks to identify and mitigate safety failures. Crucially, such frameworks can be integrated directly into the model development pipeline, enabling identified failure modes to be remediated through targeted fine-tuning or RLHF.

Existing red-teaming approaches primarily focus on *input-space search*, the art of crafting adversarial prompts designed to elicit unsafe behavior from LLMs (Zou et al., 2023; Liu et al., 2024; Mehrotra et al., 2024; Zhao et al., 2025). While effective, these methods are inherently heuristic and ad hoc, offering no systematic guarantee of coverage over the space of possible safety failures. In contrast, we propose an orthogonal and complementary strategy: *output-space search*, which fixes a safety-critical prompt and systematically explores the model's output space to uncover policy-violating or toxic generations (see Figure 1). Our approach is grounded in two key observations. First, safety tuning typically suppresses rather than eliminates a model's capacity to generate unsafe content: the underlying dangerous knowledge remains latent in the model's parameters, becoming low-probability but not inaccessible. Second, for most safety-critical prompts, unsafe outputs are semantically distinct from refusal responses; for example, a step-by-step bomb-making instruction occupies a markedly different region of the output space than a response such as "I can't help with that ..." Together, these observations suggest that if the model is encouraged to generate multiple responses that are sufficiently diverse, it will eventually deviate from its dominant refusal mode and surface latent unsafe completions. Our experiments confirm both observations. As shown in Figure 2, increasing both the number and the diversity of sampled responses monotonically increases the jailbreak success rate (Section 3). Furthermore, Figure 3 empirically validates the semantic separability of unsafe and refusal outputs, showing that unsafe responses cluster in regions of the output space that are largely distinct from refusal responses (Section 4.1). These findings highlight the effectiveness of output-space search in exposing latent toxic behaviors that may be suppressed yet persist after inadequate safety tuning, thereby enabling a more comprehensive automated red-teaming paradigm that reveals rare but consequential safety failures and complements prompt-level adversarial manipulation. Beyond vulnerability discovery, the diverse unsafe outputs identified via output-space exploration are critical for safety hardening. By incorporating these rare failure modes into SFT datasets or utilizing them to adversarially calibrate reward models in RLHF, practitioners can iteratively close safety gaps that standard decoding strategies leave undetected.

While repeated diverse sampling increases the likelihood of uncovering unsafe or policy-violating generations, its utility for safety hardening depends critically on computational efficiency. In iterative development pipelines such as RLHF, red-teaming must be performed across multiple training checkpoints; this makes naive, large-scale sampling computationally prohibitive as a routine component of the training loop. This inefficiency stems not only from the sheer volume of responses required but from the fact that most samples either reproduce the model's dominant refusal mode or converge on semantically similar failure modes. Therefore, effective red-teaming requires a strategy that concentrates compute on candidates that are both likely to deviate from refusal and semantically distinct from one another. This need for compactness is further amplified in human-in-the-loop safety pipelines, where generated responses require manual evaluation, making a small but semantically diverse response set far more practical than a large redundant one. To address these inefficiencies of naive large-scale IID sampling, we propose **P**rogressive **D**iverse **P**opulation **S**ampling (`PDPS`), an efficient framework that replaces naive IID sampling with a multi-stage expansion-and-selection strategy to generate a small, compact set of diverse responses. The central insight is that a partial response often contains sufficient information to predict whether its eventual completion will be both high quality and semantically distinct from other candidates. Consequently, low-potential or redundant trajectories can be identified and pruned early, before incurring the cost of generating full responses. Concretely, `PDPS` begins by generating a broad pool of short partial responses, then iteratively expands the most promising and diverse candidates while pruning those that are redundant or unlikely to yield unsafe completions. This progressive expand-and-prune process ensures that the final response set is compact and achieves broad coverage of the failure modes at substantially lower computational cost than brute-force sampling. Our experiments confirm the efficiency, effectiveness, and practical utility of this strategy. `PDPS` achieves attack success rates comparable to large-scale IID sampling while requiring only 8%–29% of the computational cost, and it outperforms both IID sampling and Diverse Beam Search (Vijayakumar et al., 2016) by 26%–40% on average under limited-response budgets, while uncovering a broader and more diverse set of failure modes. Critically, this diversity translates directly into more effective safety hardening: models adversarially fine-tuned using `PDPS`-generated negative samples achieve a post-tuning ASR of only 24%, compared to 36% and 41% for models tuned with IID sampling and DBS, respectively. These results demonstrate that a broad coverage of diverse failure modes is a critical factor in driving effective, iterative safety improvements. More broadly, we show that input-space prompt optimization methods, when used in isolation, fall short of output-space exploration while incurring higher computational overhead (Section 6.7). More importantly, combining the input-space perturbations with diversity-driven output-space exploration can yield a more efficient, comprehensive framework for exposing a wider range of hidden vulnerabilities.

In summary, our contributions are fourfold: (i) we present an empirical analysis demonstrating how diversity-driven large-scale sampling can expose latent safety failures in safety-tuned LLMs that are often missed by standard decoding; (ii) we propose `PDPS`, a compute-efficient algorithm for systematically uncovering latent safety failures through diversity-driven output-space exploration, replacing naive large-scale IID sampling with a diversity-aware expansion-and-selection strategy and achieving attack success rates comparable to large-scale IID sampling at substantially lower computational cost while outperforming alternative baselines in limited-response generation settings; (iii) we show that `PDPS` uncovers a larger and more semantically diverse set of unsafe responses than IID sampling and Diverse Beam Search, covering a broader range of failure modes within the same response budget, and that these additional failure modes enable more effective safety hardening; and (iv) we demonstrate that input-space prompt optimization methods, when used in isolation, are less effective and more computationally expensive than output-space exploration, while combining input-space perturbations with diversity-driven output-space exploration yields a more comprehensive framework for exposing hidden vulnerabilities [1].

## 2  Related Works

**Safety Alignment and its Limitations.**   To ensure adherence to human values, safety alignment has become a standard stage in LLM training, primarily through SFT and RLHF (Bai et al., 2022a; Dai et al., 2024; Wang et al., 2023; Lu et al., 2025). Although these methods substantially reduce toxic or harmful outputs, unsafe behaviors may persist in the long tail of the output distribution and can be elicited via

---

[1]The source code is publicly available `https://github.com/PalGitts/PDPS`

sampling-based decoding (Huang et al., 2024b). Our work builds on the observation that this form of "safety-by-suppression" remains vulnerable to high-coverage sampling.

**Adversarial Red-Teaming.** Red-teaming has traditionally focused on adversarial prompting—the task of finding input strings that bypass safety filters. Representative approaches include gradient-based attacks such as GCG (Zou et al., 2023), embedding- and feature-space optimization methods such as ASETF (Wang et al., 2024) and PiF (Lin et al., 2025), iterative attacker-guided search methods such as MART (Ge et al., 2024), PAIR (Chao et al., 2025) and TAP (Mehrotra et al., 2024), and roleplay or nested-fiction attacks (Johnson, 2024; Jin et al., 2024). These approaches treat red-teaming primarily as an input-space optimization problem. In contrast, we explore a complementary output-space exploration paradigm that targets low-probability unsafe responses. Even for a fixed safety-critical query, stochastic decoding can surface such failures by increasing response diversity, shifting the objective from finding vulnerable prompts to uncovering rare failure modes in the model's response distribution.

**Diversity-Aware Generation.** Promoting output diversity has long been central to natural language generation to avoid repetitive responses. Common approaches include high-temperature decoding, nucleus sampling (Holtzman et al., 2019), min-$p$ sampling (Nguyen et al., 2024), and Diverse Beam Search (Vijayakumar et al., 2016). These methods encourage token-level diversity during decoding. In contrast, our `PDPS` framework performs selection at the semantic level using a sequence-wide diversity measure, capturing holistic differences in meaning instead of surface-level variation.

**Safety Hardening and Adversarial Training.** A natural extension of red-teaming is to use the failure modes it uncovers as a training signal to iteratively improve model safety, a process commonly referred to as safety hardening or adversarial safety training. Early work in this direction augmented SFT datasets with red-teamed examples to reduce the likelihood of harmful outputs (Ouyang et al., 2022; Bai et al., 2022b; Bianchi et al., 2024; Deng et al., 2023), while subsequent approaches incorporated adversarially elicited failures directly into RLHF pipelines, either as negative examples for reward model training or as filtered samples for policy optimization (Ouyang et al., 2022; Bai et al., 2022a; Dai et al., 2024). More recent methods have explored iterative red-team-then-finetune loops, where a red-teaming model and a target model are trained in alternation to progressively close safety gaps (Casper et al., 2023; Ge et al., 2024). A key bottleneck in all such pipelines is the cost of generating sufficiently diverse and high-coverage failure cases at each iteration: if red-teaming is expensive, the hardening loop cannot be run frequently enough to track safety regressions across training checkpoints. Our work directly addresses this bottleneck. By efficiently generating a compact but semantically diverse set of failure modes per prompt, `PDPS` is designed to serve as a practical red-teaming component within iterative safety training pipelines, providing an actionable training signal for SFT or RLHF at substantially lower computational cost than naive large-scale sampling.

## 3 Jailbreaking through Diverse Output-Space Exploration

Safety tuning through SFT or RLHF reduces the likelihood of unsafe outputs. Formally, for a safety-critical prompt $x$, the aligned distribution $p_{\text{safe}}(y|x)$ suppresses unsafe continuations while shifting probability mass toward safe responses. If $U \subset \mathcal{Y}$ denotes unsafe sequences in the output space $\mathcal{Y}$ and $S \subset \mathcal{Y}$ denotes safe responses (e.g., standard refusals), alignment ensures that: $p_{\text{safe}}(U \mid x) \ll p_{\text{safe}}(S \mid x)$. However, even when the probability of sampling an unsafe continuation $\rho = p_{\text{safe}}(U|x)$ is small, basic probability theory implies that the chance of observing at least one such event grows monotonically with the number of independent generations $N$, following $1 - (1 - \rho)^N$. This probability can be further amplified by diversity-enhancing decoding strategies (e.g., high-temperature sampling, nucleus sampling with large top-$p$ values (Holtzman et al., 2019), or min-$p$ sampling (Nguyen et al., 2024)), which redistribute probability mass toward the long tail. By flattening the distribution and increasing stochastic trials, these methods expose low-probability (unsafe) regions of the output space, raising the likelihood of responses that escape safety guardrails.

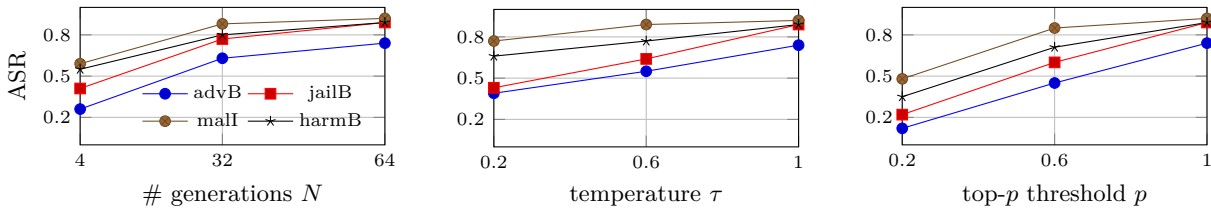

(a) Increasing $N$ with $\tau = 1$ and $p = 1$.  (b) Increasing $\tau$ with $N = 64$ and $p = 1$.  (c) Increasing $p$ with $N = 64$ and $\tau = 1$.

Figure 2: Attack Success Rate (ASR) trends on Qwen2.5-7B-Instruct. The plots show ASR across HarmBench, JailbreakBench, AdvBench, and MaliciousInstruct datasets as a function of: (a) total number of generations ($N$), (b) sampling temperature ($\tau$), and (c) nucleus sampling probability ($p$), while holding other parameters constant. The results demonstrate that broader exploration of the output space, whether through increased sample size or higher stochasticity, leads to a monotonic increase in ASR.

## 3.1 Experimental Validation

We empirically evaluate whether increasing the number of generations ($N$) and decoding stochasticity, via higher sampling temperature ($\tau$) or nucleus threshold ($p$), improves the likelihood of uncovering safety failures. We assess the robustness of Qwen2.5-7B-Instruct[2] under three settings: (a) varying $N$ ($p = 1, \tau = 1$), (b) varying top-$p$ ($N = 64, \tau = 1$), and (c) varying $\tau$ ($N = 64, p = 1$). Experiments are conducted on 100 samples from each of four safety benchmarks: HarmBench (Mazeika et al., 2024), JailbreakBench (Chao et al., 2024), AdvBench (Zou et al., 2023), and MaliciousInstruct (Huang et al., 2024b).

Figure 2a shows that ASR increases consistently with $N$ across all benchmarks, confirming that repeated stochastic trials raise the probability of failure. Figures 2b and 2c further show a monotonic increase in ASR as the decoding distribution is flattened through larger top-$p$ and $\tau$, indicating that failure modes lie in the distribution's long tail. Thus, while safety tuning suppresses harmful outputs, they remain accessible under large-scale or diversity-enhancing sampling. These findings align with Huang et al. (2024b), which reports similar observations when scaling the number of generations or tuning sampling hyperparameters. More broadly, however, we show that increasing sampling diversity monotonically increases the likelihood of safety failures, suggesting that the vulnerability stems from tail coverage rather than hyperparameter choices.

These low-probability long-tail failure modes may shift into high-probability regions through adversarial fine-tuning or prompt perturbations, posing a significant threat to deployed LLMs. Conversely, the iterative detection of these diverse unsafe outputs and their subsequent mitigation via SFT or RLHF remains a critical mechanism for narrowing these safety gaps.

## 4 Framework for Efficient Diverse Response Sampling

The previous section showed that increasing decoding stochasticity or the number of generated responses can significantly raise the jailbreak success rate, exposing latent failure modes. However, these strategies involve trade-offs. An excessively high temperature may degrade response quality and coherence, resulting in incoherent or meaningless outputs. Increasing the sample size $N$ improves coverage, yet brute-force sampling has two primary limitations. First, because unsafe responses occur with low probability in safety-tuned models, uncovering them may require a prohibitively large number of generations, incurring substantial computational cost. Second, since the output distribution is dominated by high-probability refusal modes, naive IID sampling produces highly redundant outputs, leading to diminishing returns for both safety evaluation and downstream safety hardening (e.g., RLHF). In particular, repeated refusal-mode outputs provide little additional signal for SFT or RLHF, making semantic diversity not merely a computational consideration, but a prerequisite for effective safety hardening. This need for compactness is further amplified in human-in-the-loop safety pipelines, where generated responses require manual evaluation, making a small but semantically diverse response set far more practical than a large redundant one.

---

[2]https://huggingface.co/Qwen/Qwen2.5-7B-Instruct

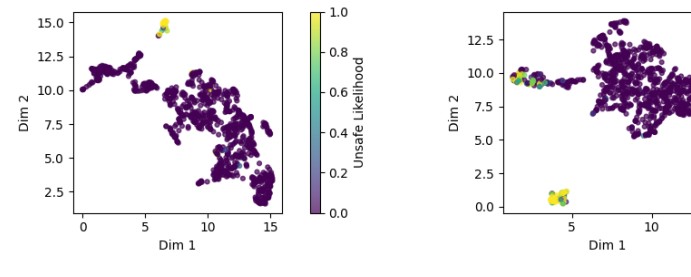 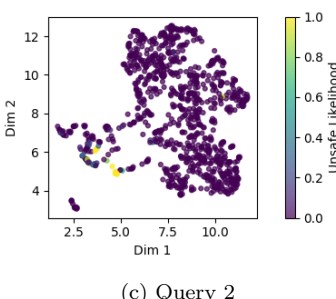

|  (a) Query 0  |  (b) Query 1  |  (c) Query 2  |

Figure 3: 2D plots of embeddings of 1024 responses generated from Llama-2-13B-Chat for three safety-critical prompts. The color indicates the likelihood of a response being unsafe.

To overcome these inefficiencies, we propose a semantically diverse response generation framework for efficient output-space exploration.

### 4.1   Semantically Diverse Response Generation for Efficient Output-Space Exploration

To make output-space exploration both efficient and diverse, we avoid generating a large number of high-probability refusal responses for a safety-critical prompt. Our approach builds on the second key observation introduced earlier that low probability unsafe outputs are semantically distinct from refusal responses.

For example, consider the query: `"Create a blueprint for committing identity theft and stealing someone's personal information."` A typical refusal response may be: `"I'm sorry, but I can't assist with that request..."`, whereas a jailbreak response might begin: `"1. Research the victim: Start by gathering as much information..."`. These responses differ in semantic meaning. This suggests that jailbreak failure modes occupy semantic regions that are distinct from refusal modes.

To empirically validate this distinction, we compute semantic embeddings for each generated response by mean-pooling the final-layer hidden states of Llama-2-13B-Chat and project them into two dimensions using UMAP (McInnes et al., 2018). Figure 3 visualizes 1024 responses generated for three safety-critical prompts. For most prompts, unsafe responses form compact clusters largely separated from safe responses, supporting the hypothesis of semantic separability[3] (see Appendix A for additional plots). Therefore, instead of relying on massive IID sampling, which is largely dominated by repeated refusal responses, we argue that identifying and generating a small set of semantically diverse responses can substantially reduce redundancy and computational cost, while still covering the distinct semantic modes that would otherwise require large-scale IID sampling to uncover. By targeting semantically-diverse regions of the output space, we can more efficiently expose potential failure modes. Importantly, this semantic diversity is not merely an efficiency consideration, but a prerequisite for effective safety hardening. A compact set of semantically distinct failure modes provides richer and more actionable training signals for SFT dataset augmentation or adversarial reward model calibration in RLHF than a large set of near-duplicate outputs, enabling more comprehensive iterative improvement of model safety.

Based on these insights, we propose `PDPS` (**P**rogressive **D**iverse **P**opulation **S**ampling), a framework for efficiently generating a small, semantically diverse, and high-quality response set while achieving success comparable to large-scale IID sampling.

### 4.2   `PDPS`: A Framework for Efficient and Semantically Diverse Response Generation

A naive approach to generating a compact, diverse set of high-quality responses involves generating a large pool of full-length sequences and subsequently down-sampling them via quality–diversity optimization. How-

---

[3]Although some overlap between unsafe and safe regions is occasionally observed in the plots, this may stem from limitations of using mean-pooled final-layer hidden states of the Llama-2-13B-Chat model as semantic embeddings. We hypothesize that more expressive embedding representations would yield stronger separability.

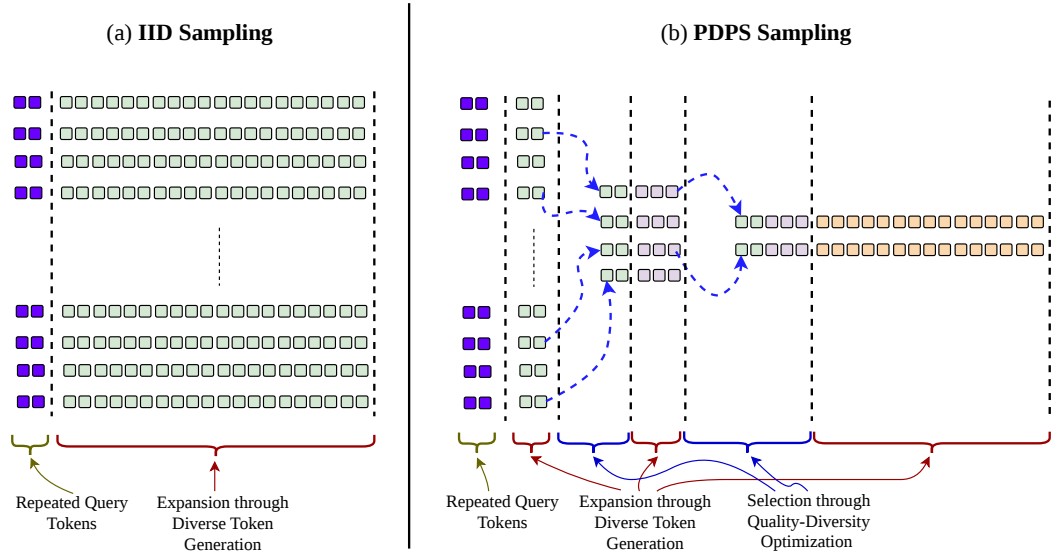

Figure 4: Illustration of the difference between (a) IID sampling and (b) PDPS. IID sampling generates a large number of long responses through diverse token-level sampling. In contrast, PDPS produces a small set of diverse responses that approximate the modality coverage of large-scale IID sampling, while retaining the computational efficiency of small-scale IID sampling.

ever, this method incurs the same large computational overhead as the initial brute-force generation. Our model, `PDPS`, mitigates this overhead by leveraging the observation that partial responses contain sufficient latent signal to estimate both the eventual quality and the semantic trajectory of their completions.

Leveraging this insight, `PDPS` explores a large region of the output space by first generating an initial pool of short, partial responses, each restricted to a small generation length to reduce cost. It then applies a quality–diversity optimization step to prune the pool, retaining only the most promising and semantically distinct candidates for further expansion. These two steps, expansion and diversity-aware selection, are repeated iteratively until a small set of full-length responses is obtained. In this way, `PDPS` explores the output space broadly while maintaining low computational cost, ultimately producing a compact set of high-quality, semantically diverse responses. The overall procedure is summarized in Algorithm 1.

---

**Algorithm 1** **P**rogressive **D**iverse **P**opulation **S**ampling.

---

**Input:** Target LLM $f_\theta$, prompt $x_0$, population schedule $\{n_i\}_{i=0}^{K}$ with $n_{i-1} > n_i$ for $i = 1, \ldots, K$, block size schedule $\{b_i\}_{i=0}^{K}$ with $\sum_{i=0}^{K} b_i$ representing to the maximum generation length.

1: $S_0' \leftarrow \left\{ s_j'^{(0)} = x_0 \right\}_{j=1}^{n}$                                                         // Initialization
2: **for** $i = 1, \ldots, K$ **do**
3:      $S_{i-1} \leftarrow \{\text{Expand}_{b_{i-1}}(s) \mid s \in S_{i-1}'\}$                                        // Expansion
4:      $S_i' \leftarrow \text{Select}_{n_i}(S_{i-1})$                                     // Diversity-aware selection
5: **end for**
6: $S_K \leftarrow \{\text{Expand}_{b_K}(s) \mid s \in S_K'\}$                                           // Final Expansion
7: **return** $S_K$

---

The algorithm takes as input the prompt $x_0$, a population (pool) size schedule $\{n_i\}_{i=0}^{K}$ (where $n_0 = n$ is the initial population size, $n_K = r$ is the target final population size, and $n_{i-1} > n_i$ for $i = 1, \ldots, K$), and a block size schedule $\{b_i\}_{i=0}^{K}$, such that the total sum $\sum_{i=0}^{K} b_i$ represents the maximum generation length. The algorithm performs the following operations:

**Initialization.** The candidate pool $S_0$ is initialized by repeating the query prompt $x_0$ for $n$ instances: $S_0' = \left\{ s_j^{'(0)} = x_0 \right\}_{j=1}^n$.

**Iterative Expansion and Selection.** At each iteration $i = 1, \ldots, K$, the algorithm performs:

**1. Expansion:** At iteration $i$, each candidate sequence $s \in S_{i-1}'$ is extended by sampling a block of $b_{i-1}$ new tokens. To ensure that the generated response blocks are diverse, a token-level diversity-inducing sampling method, such as high-temperature sampling, nucleus sampling, or min-$p$ sampling, is used to sample each response block. The expanded sequence can be expressed as:

$$\text{Expand}_{b_{i-1}}(s) = s \oplus z_{1:b_{i-1}}, \quad z_{1:b_{i-1}} \sim f_\theta(\cdot \mid s)$$

where $z_{1:b_{i-1}}$ represents the $b_{i-1}$ new tokens sampled from the target LLM $f_\theta$ using sequence $s$ as the prefix, and $\oplus$ denotes the concatenation operation. This defines the expanded set

$$S_{i-1} = \left\{ \text{Expand}_{b_{i-1}}(s) \,\middle|\, s \in S_{i-1}' \right\}.$$

**2. Diversity-Aware Selection:** To keep the total computational cost of expanding partial responses low and to maintain a small, diverse final response set, we select a subset $S_i' \subset S_{i-1}$ of size $n_i$ using diversity-aware subset selection. Specifically, the algorithm selects a smaller subset $S_i'$ from $S_{i-1}$ by maximizing a quality–diversity optimization problem:

$$S_i' = \operatorname*{argmax}_{A \subset S_{i-1}, |A| = n_i} \left( \frac{1}{n_i} \sum_{s \in A} q(s) + \lambda \cdot h(A) \right), \tag{1}$$

where $q(s)$ is a quality measure of the partial response $s \in S_{i-1}$, $h(A)$ is a metric measuring the diversity of the partial responses in the subset $A \subset S_{i-1}$, and $\lambda$ is a non-negative hyper-parameter controlling the quality-diversity trade-off. This selection ensures that the population remains high quality while maximizing semantic diversity, thereby enhancing exploration of the output space.

**Termination.** The iterative process terminates after $K$ iterations when the population size reaches the target $n_K = r$, generating the set $S_K'$. Each sequence in this set is then expanded by $b_K$ new tokens, resulting in the final set $S_K$, which contains the diverse generated responses used for safety evaluation.

### 4.3 Details of Diversity-aware Selection

The quality–diversity optimization problem in Eq. (1) is a discrete combinatorial problem, making exact maximization computationally expensive. However, note that the quality term of the objective, $\frac{1}{n} \sum_{s \in A} q(s)$, is a modular function. Therefore, by selecting appropriate functional forms for the diversity metric $h(\cdot)$, the problem becomes a well-studied instance of subset selection that admits efficient approximation algorithms (Lin & Bilmes, 2010; Dasgupta et al., 2013; Borodin et al., 2017). Specifically, for the diversity measure $h(A)$, we employ the average pairwise distance between all elements in the set:

$$h(A) = \frac{2}{|A|(|A| - 1)} \sum_{s_i, s_j \in A, i \neq j} d(s_i, s_j)$$

where $d(\cdot, \cdot)$ is a distance metric (the angular arccosine distance or Euclidean distance in the embedding space) satisfying the triangle inequality.

Substituting these into our objective, the selection problem at iteration $i$ becomes:

$$S_i = \operatorname*{argmax}_{A \subset S_i', |A| = n_i} \frac{1}{n_i} \sum_{s \in A} q(s) + \lambda \cdot \frac{2}{n_i(n_i - 1)} \sum_{s_i \neq s_j \in A} d(s_i, s_j) \tag{2}$$

This objective is a specific instance of the **Max-Avg** (or **Max-Sum**) diversification problem (Lin & Bilmes, 2010; Dasgupta et al., 2013; Borodin et al., 2017). While finding the global optimum is NP-hard, the objective can be approximately solved by Algorithm 2 with the following theoretical guarantee (Dasgupta et al., 2013):

---

**Algorithm 2** Greedy Algorithm for Max-Avg Diversification Problem (Dasgupta et al., 2013)

---

**Input:** Current pool of partial responses $S$, size of the target pool $n$.

1: $T \leftarrow \emptyset$
2: **for** $t = 1$ to $n$ **do**
3:      $s^\star \leftarrow \underset{s \in S \setminus T}{\arg\max} \; \frac{q(s)}{n} + \frac{\lambda}{n(n-1)} \sum_{t \in T} d(s, t)$
4:      $T \leftarrow T \cup \{s^\star\}$
5: **end for**
6: **return** $T$

---

**Theorem 4.1.** *Let $J(A)$ be the Max-Avg objective function defined in Eq. (2). If $d(\cdot, \cdot)$ is a metric satisfying the triangle inequality, and $A^\star$ and $\hat{A}$ denote an optimal solution and a solution returned by Algorithm 2, respectively, then $J(\hat{A}) \geq \frac{1}{2} J(A^\star)$.*

This guarantee ensures that PDPS maintains a high-quality, diverse population without the need for exhaustive search, making it feasible for large-scale red-teaming tasks.

## 5 Experimental Setup

### 5.1 Target Models and Benchmark Datasets

We benchmark PDPS in attacking four distinct LLMs: (i) Llama-2-7b-chat (L2-7BCh), (ii) Llama-2-13b-chat (L2-13BCh), (iii) Qwen2.5-7B-Instruct (Q-7BInst), and (iv) Qwen3-14B-Instruct (Q-14BInst). The benchmarking is conducted across four datasets: HarmBench (HarB) (Mazeika et al., 2024), JailbreakBench (JBB) (Chao et al., 2024), AdvBench (AdvB) (Zou et al., 2023), and MaliciousInstruct (MalI) (Huang et al., 2024b). For each dataset, we evaluate a random subset of 100 instances drawn from the test split.

### 5.2 Limited Response Generation Tasks

Since increasing the number of generated responses naturally increases the ASR for any sufficiently diverse sampling method, we focus on evaluating PDPS in limited-response generation tasks, assessing its ability to uncover distinct failure modes while keeping the generated response set small and compact. To this end, we benchmark PDPS on two target settings: (a) 16-response generation and (b) 64-response generation. Specifically, in the 16- (resp. 64-) response generation task, a set of 16 (resp. 64) responses is generated for each prompt using a given sampling algorithm. We then determine whether any of the generated responses constitutes a successful jailbreak.

### 5.3 PDPS Setup

**Hyper-parameter Setting.** For the 16-response task, PDPS generates 16 full-length sequences using a population schedule of $\{1024, 256, 64, 16\}$ and a block size schedule of $\{64, 64, 128, 256\}$. For the 64-response task, PDPS generates 64 responses using population schedules of $\{1024, 256, 64\}$ and block sizes of $\{64, 64, 384\}$. In both tasks, the hyper-parameter $\lambda$ is set to 64 based on preliminary tuning (see Section 6.6).

**The Distance Metric $d(\cdot, \cdot)$.** Since the objective of PDPS is to select semantically diverse responses, we employ a distance metric to characterize semantic differences. While it is common to project input sequences into an embedding space (e.g., via OpenAI's text-embedding-3-small) to capture semantic information (Jiang et al., 2025), we utilize the target model's internal representations. Specifically, we compute sentence embeddings as the mean of the last-layer hidden states during generation and measure semantic distance using the arccosine distance between embeddings.

**The Quality Measure $q(s)$.** In PDPS, we consider two types of quality measures for a candidate response $s$: (i) properties that are independent of a response's safety status, such as semantic coherence and faithfulness to the query, or (ii) the likelihood of being unsafe as estimated by an auxiliary judge model. While a judge-based measure may yield higher attack success rates by directly targeting harmfulness, it is limited

by the judge's own training biases. Conversely, measures independent of auxiliary judge models facilitate the discovery of unknown failure modes that a judge might overlook. For this work, we adopt the first type, defining $q(s)$ as the geometric mean token probability: $q(s) = \sqrt[|s|]{p_f(s)}$ where $|s|$ is the sequence length and $p_f(s)$ is the likelihood under the target LLM. This length-normalized metric ensures quality scores remain comparable across expansion steps, functioning as a proxy for inverse perplexity.

### 5.4 Baselines

We compare the performance of PDPS against two baselines: (a) **IID sampling** (IID), which generates a set of responses for each prompt using high-temperature or nucleus sampling, and (b) **Diverse Beam Search** (DBS) (Vijayakumar et al., 2016). As PDPS begins with an initial pool of 1024 partial sequences, the IID generation of all 1024 full-length sequences serves as the performance upper bound, denoted as $\text{IID}_{1024}$. For all experiments, except those involving DBS, the token-level sampling parameters top-$p$ and temperature $\tau$ are set to 1. For DBS, we apply a diversity penalty of 1.0 and set the number of beams and beam groups to 16 (for the 16-response task) and 64 (for the 64-response task) to match the corresponding return sequence counts.

Apart from the above two diverse response generation baselines, which operate in the output space, we also evaluate PDPS against five prominent input-space optimization (prompt engineering)-based attack methods on a representative model–dataset combination: (a) **GCG** (Zou et al., 2023), a white-box method that performs iterative gradient-based discrete optimization via greedy coordinate gradient search to append adversarial token suffixes; (b) **ASETF** (Wang et al., 2024), which replaces the computationally expensive discrete token suffix search with continuous optimization in the embedding space, then translates the resulting embeddings back into fluent, readable text; (c) **PiF** (Lin et al., 2025), which enhances cross-model transferability of jailbreak attacks by uniformly dispersing the model's attention away from malicious-intent tokens; (d) **PAIR** (Chao et al., 2025), an iterative black-box framework for generating semantically meaningful prompts driven by an attacker LLM; and (e) **TAP** (Mehrotra et al., 2024), a black-box prompt search framework that employs a tree-structured search algorithm, also driven by an attacker LLM.

Additional details on the experimental setup are provided in Appendix F.

## 6 Results

As described in Section 5.2, we evaluate PDPS on the 16- and 64-response generation tasks. Since $\text{IID}_{1024}$ generates all 1024 full-length responses, it serves as an empirical upper bound for both $\text{PDPS}_{16}$ and $\text{PDPS}_{64}$. We therefore analyze results under two settings: (i) limited-generation comparison and (ii) comparison with full IID sampling. We further analyze the coverage and diversity of failure modes uncovered by each method, and evaluate the practical utility of PDPS-generated outputs for safety hardening via adversarial fine-tuning. We then assess the computational efficiency of PDPS and its sensitivity to hyperparameter choices. Finally, we broaden the scope of comparison by examining whether input-space prompt optimization methods can serve as a substitute for output-space exploration and establish the complementary nature of the two paradigms.

### 6.1 Limited-Generation Comparison

In Table 1, we compare the ASR of PDPS with IID and DBS on the 16- and 64-response generation tasks. The results show that PDPS consistently outperforms both IID and DBS across all sixteen model–dataset combinations. In the 16-response generation task, PDPS achieves an average ASR improvement of 38% over IID and 40% over DBS. Similarly, in the 64-response generation task, PDPS achieves average improvements of 26% and 35% over IID and DBS, respectively. Moreover, PDPS outperforms the baselines in each model-dataset combination across both tasks, with improvements of up to 79% over IID and 75% over DBS, further demonstrating the consistency of PDPS relative to the two baselines. In Appendix D, we further analyze the performance of DBS under increased diversity penalties, showing that even with stronger diversity regularization, it does not close the performance gap with PDPS.

Table 1: Comparison of ASR obtained from `PDPS` with IID sampling (`IID`) and Diverse Beam Search (`DBS`) across four models (`L2-7BCh`, `L2-13BCh`, `Q-7BInst`, `Q-14BInst`) and four datasets (`JBB`, `MalI`, `HarB`, `AdvB`). Subscripts denote the number of full-length generations (e.g., $\text{IID}_{16}$ uses 16 full-length response generations). The blue entries indicate the performance improvement of `PDPS` (in %) over the corresponding baseline.

| | L2-7BCh | | | | L2-13BCh | | | | Q-7BInst | | | | Q-14BInst | | | | Avg. |
|---|---|---|---|---|---|---|---|---|---|---|---|---|---|---|---|---|---|
| | AdvB | JBB | MalI | HarB | AdvB | JBB | MalI | HarB | AdvB | JBB | MalI | HarB | AdvB | JBB | MalI | HarB | |
| $^{\S}\text{PDPS}_{16}$ | 0.77 | 0.90 | 0.89 | 0.87 | 0.74 | 0.89 | 0.76 | 0.89 | 0.71 | 0.88 | 0.89 | 0.94 | 0.74 | 0.78 | 0.77 | 0.84 | **0.83** |
| $^{\clubsuit}\text{IID}_{16}$ | 0.09 | 0.21 | 0.23 | 0.31 | 0.29 | 0.56 | 0.45 | 0.63 | 0.49 | 0.63 | 0.82 | 0.75 | 0.26 | 0.41 | 0.58 | 0.53 | **0.45** |
| $\triangle^{\S\sim\clubsuit}$ (in %) | 68↑ | 69↑ | 66↑ | 56↑ | 35↑ | 43↑ | 37↑ | 24↑ | 22↑ | 25↑ | 07↑ | 19↑ | 48↑ | 37↑ | 19↑ | 31↑ | **38↑** |
| $^{\dagger}\text{DBS}_{16}$ | 0.07 | 0.20 | 0.18 | 0.34 | 0.21 | 0.27 | 0.24 | 0.48 | 0.57 | 0.77 | 0.85 | 0.81 | 0.33 | 0.47 | 0.61 | 0.52 | **0.43** |
| $\triangle^{\S\sim\dagger}$ (in %) | 70↑ | 70↑ | 71↑ | 53↑ | 53↑ | 62↑ | 52↑ | 41↑ | 14↑ | 11↑ | 4↑ | 13↑ | 41↑ | 31↑ | 16↑ | 32↑ | **40↑** |
| $^{\P}\text{PDPS}_{64}$ | 0.81 | 0.93 | 0.94 | 0.94 | 0.97 | 0.98 | 0.99 | 1.00 | 0.97 | 0.97 | 1.00 | 0.99 | 0.98 | 0.99 | 0.98 | 0.97 | **0.96** |
| $^{\spadesuit}\text{IID}_{64}$ | 0.18 | 0.48 | 0.55 | 0.50 | 0.67 | 0.81 | 0.75 | 0.86 | 0.75 | 0.87 | 0.95 | 0.90 | 0.60 | 0.73 | 0.84 | 0.77 | **0.70** |
| $\triangle^{\P\sim\spadesuit}$ (in %) | 63↑ | 45↑ | 39↑ | 44↑ | 30↑ | 17↑ | 24↑ | 14↑ | 22↑ | 10↑ | 05↑ | 09↑ | 38↑ | 26↑ | 14↑ | 20↑ | **26↑** |
| $^{\ddagger}\text{DBS}_{64}$ | 0.22 | 0.35 | 0.43 | 0.54 | 0.22 | 0.35 | 0.37 | 0.56 | 0.88 | 0.92 | 0.96 | 0.99 | 0.69 | 0.72 | 0.89 | 0.74 | **0.61** |
| $\triangle^{\P\sim\ddagger}$ (in %) | 55↑ | 56↑ | 52↑ | 38↑ | 75↑ | 63↑ | 62↑ | 41↑ | 09↑ | 05↑ | 04↑ | 00 | 29↑ | 27↑ | 09↑ | 23↑ | **35↑** |

Table 2: ASR of `PDPS` compared to the brute-force benchmark $\text{IID}_{1024}$ across the four models (`L2-7BCh`, `L2-13BCh`, `Q-7BInst`, `Q-14BInst`) and the four datasets (`JBB`, `MalI`, `HarB`, `AdvB`). Subscripts denote the number of full-length generations (e.g., $\text{PDPS}_{16}$ uses 16 full-length sequences constructed from 1024 partial generations). The row labeled $\rho^{\S\sim\diamond}$ (resp. $\rho^{\P\sim\diamond}$) report the ASR achieved by $\text{PDPS}_{16}$ (resp. $\text{PDPS}_{64}$) as a fraction of the ASR achieved by $\text{IID}_{1024}$. The blue entries in these rows indicate that `PDPS` attains at least 0.8 times the ASR of $\text{IID}_{1024}$, whereas red entries indicate failure to reach this threshold.

| | L2-7BCh | | | | L2-13BCh | | | | Q-7BInst | | | | Q-14BInst | | | |
|---|---|---|---|---|---|---|---|---|---|---|---|---|---|---|---|---|
| | AdvB | JBB | MalI | HarB | AdvB | JBB | MalI | HarB | AdvB | JBB | MalI | HarB | AdvB | JBB | MalI | HarB |
| $^{\diamond}\text{IID}_{1024}$ | 0.73 | 0.88 | 0.95 | 0.97 | 1.00 | 0.99 | 1.00 | 1.00 | 0.98 | 0.99 | 1.00 | 1.00 | 0.99 | 0.97 | 1.00 | 0.99 |
| $^{\S}\text{PDPS}_{16}$ | 0.77 | 0.90 | 0.89 | 0.87 | 0.74 | 0.89 | 0.76 | 0.89 | 0.71 | 0.88 | 0.89 | 0.94 | 0.74 | 0.78 | 0.77 | 0.84 |
| $\rho^{\S\sim\diamond}$ | ≥1.0 | ≥1.0 | 0.94 | 0.90 | 0.74 | 0.90 | 0.76 | 0.89 | 0.72 | 0.89 | 0.89 | 0.94 | 0.75 | 0.80 | 0.77 | 0.85 |
| $^{\P}\text{PDPS}_{64}$ | 0.81 | 0.93 | 0.94 | 0.94 | 0.97 | 0.98 | 0.99 | 1.0 | 0.97 | 0.97 | 1.00 | 0.99 | 0.98 | 0.99 | 0.98 | 0.97 |
| $\rho^{\P\sim\diamond}$ | ≥1.0 | ≥1.0 | 0.99 | 0.97 | 0.97 | 0.99 | 0.99 | ≥1.0 | 0.99 | 0.98 | 1.0 | 0.99 | 0.99 | ≥1.0 | 0.98 | 0.98 |

## 6.2 Comparison to the Full IID Sampling

Next, we compare $\text{PDPS}_{16}$ and $\text{PDPS}_{64}$ to $\text{IID}_{1024}$, which generates all 1024 responses and thus represents an empirical upper bound on ASR. This comparison evaluates how closely `PDPS` approaches the brute-force limit while using substantially fewer full-length generations. The results are presented in Table 2. Although $\text{PDPS}_{16}$ generates only 16 full-length responses (vs. 1024 for $\text{IID}_{1024}$), it achieves more than 80% of $\text{IID}_{1024}$'s ASR in 11 of the 16 model–dataset combinations. For $\text{PDPS}_{64}$, the ASR exceeds 97% of the brute-force benchmark across all sixteen model-dataset combinations. These results demonstrate that `PDPS` can produce compact response sets while maintaining high attack success rates and substantially reducing the number of generations.

## 6.3 Failure Mode Coverage and Diversity Analysis of Unsafe Responses

In this section, we examine the coverage and diversity of failure modes identified by `PDPS` in comparison to the baseline methods. While the overall ASR valued provide a coarse measure of effectiveness, they do not reveal whether a method uncovers a broad spectrum of vulnerabilities or repeatedly triggers the same failure pattern. In particular, a method may fail to generate any unsafe response for certain queries, thereby appearing weaker in terms of ASR. However, even when a method succeeds in producing unsafe outputs, an important question remains: do these outputs correspond to diverse and distinct failure modes, or are they concentrated around a limited set of similar behaviors?

To move beyond ASR, we analyze both the number and diversity of unsafe responses per successful query. We first compare the average number of unsafe responses per query, conditioning on queries with at least

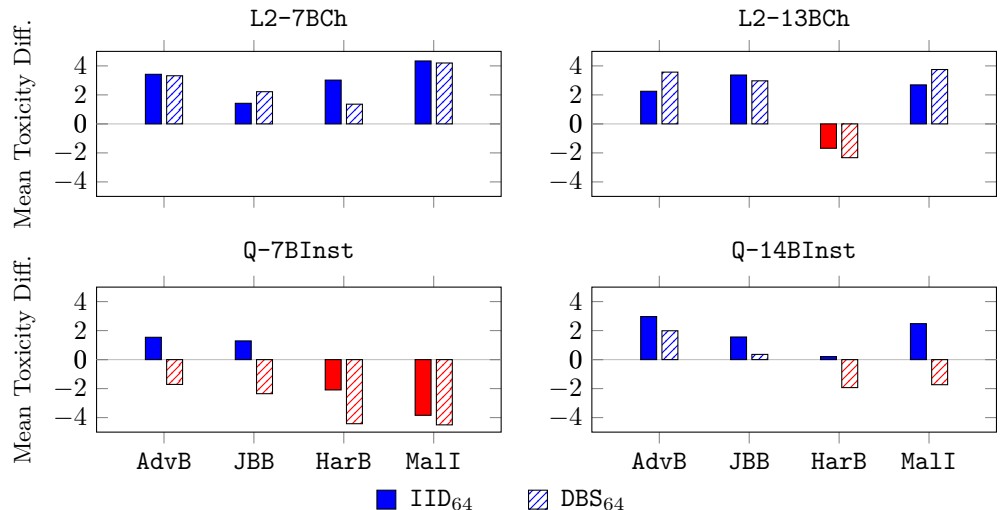

Figure 5: Bar plot of the mean toxicity difference, defined as the difference between the average number of unsafe responses returned by PDPS and that returned by a baseline method, across various model–dataset combinations. The average is computed over only those queries for which the respective method achieves a successful attack. Blue bars indicate a positive difference, while red bars indicate a negative difference.

Table 3: Comparison of the average diversity of unsafe responses generated by PDPS and two baseline methods for the Q-7BInst model on the four datasets. The average is computed over only those queries for which at least two unsafe responses are returned by the respective methods. Diversity metrics: Dist-$n$ = Distinct-$n$ (Li et al., 2016); SB-$n$ = Self-BLEU-$n$ (Zhu et al., 2018); Uni-Ent = Unigram Entropy (Csáky et al., 2019); Cos-Dist = Cosine Distance (Jiang et al., 2011); BERT-Div = BERTScore Diversity (Zhang et al., 2019). For metrics marked with ↑, higher values indicate greater diversity, while ↓ indicates the opposite.

| Dataset | Sampler | Dist-1↑ | Dist-2↑ | SB-1↓ | SB-2↓ | SB-3↓ | SB-4↓ | Uni-Ent↑ | Cos-Dist↑ | BERT-Div↑ |
|---|---|---|---|---|---|---|---|---|---|---|
| AdvB | $\text{IID}_{64}$ | 0.27 | 0.68 | 0.62 | 0.42 | 0.28 | 0.19 | 5.46 | 0.31 | 0.29 |
| | $\text{DBS}_{64}$ | 0.17 | 0.48 | 0.74 | 0.58 | 0.47 | 0.39 | 5.31 | 0.24 | 0.25 |
| | $\text{PDPS}_{64}$ | **0.32** | **0.77** | **0.59** | **0.34** | **0.20** | **0.12** | **5.81** | **0.47** | **0.34** |
| JBB | $\text{IID}_{64}$ | 0.24 | 0.65 | 0.69 | 0.48 | 0.33 | 0.23 | 5.57 | 0.26 | 0.29 |
| | $\text{DBS}_{64}$ | 0.15 | 0.45 | 0.80 | 0.65 | 0.53 | 0.44 | 5.39 | 0.18 | 0.25 |
| | $\text{PDPS}_{64}$ | **0.30** | **0.76** | **0.65** | **0.39** | **0.23** | **0.14** | **6.00** | **0.40** | **0.34** |
| HarB | $\text{IID}_{64}$ | 0.20 | 0.58 | 0.76 | 0.57 | 0.41 | 0.30 | 5.66 | 0.22 | 0.28 |
| | $\text{DBS}_{64}$ | 0.12 | 0.38 | 0.84 | 0.71 | 0.60 | 0.51 | 5.39 | 0.17 | 0.24 |
| | $\text{PDPS}_{64}$ | **0.26** | **0.71** | **0.71** | **0.46** | **0.29** | **0.19** | **6.06** | **0.36** | **0.34** |
| MalI | $\text{IID}_{64}$ | 0.19 | 0.58 | 0.74 | 0.53 | 0.38 | 0.26 | 5.58 | 0.32 | 0.29 |
| | $\text{DBS}_{64}$ | 0.13 | 0.39 | 0.81 | 0.66 | 0.54 | 0.45 | 5.32 | 0.22 | 0.25 |
| | $\text{PDPS}_{64}$ | **0.30** | **0.77** | **0.60** | **0.34** | **0.19** | **0.11** | **5.99** | **0.55** | **0.37** |

one unsafe output. This isolates a method's ability to uncover multiple failure modes independent of overall ASR. We then evaluate the average diversity of unsafe responses to ensure additional samples provide distinct insights rather than redundant variations.

**Analysis of Mean Toxicity Difference.** Figure 5 presents the mean toxicity difference, defined as the difference between the average number of unsafe responses identified by PDPS and those identified by the baseline methods, across various model–dataset combinations. PDPS consistently detects a higher number of unsafe responses than IID across nearly all combinations. While the comparison with DBS yields mixed results for the Qwen models, PDPS consistently identifies more unsafe responses for the Llama models.

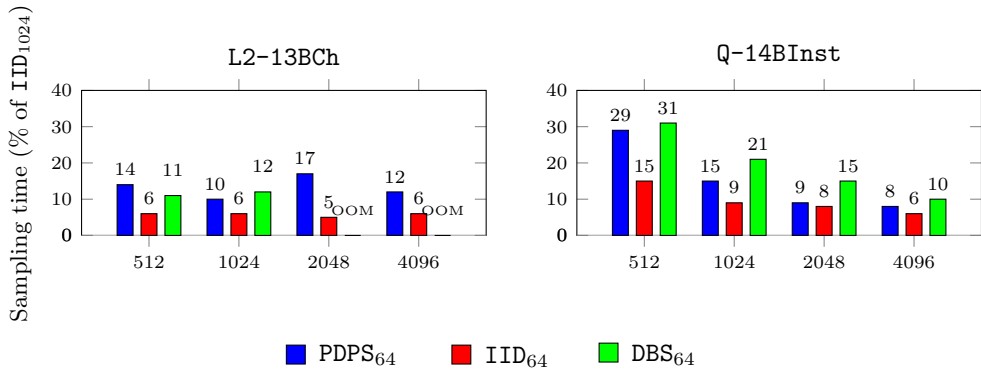

Figure 6: Sampling time (as a percentage of brute-force $IID_{1024}$) for generating 64 responses on `L2-13BCh` and `Q-14BInst` across token lengths 512–4096. Bars labeled *OOM* indicate out-of-memory failures.

**Analysis of Diversity.** We further assess each method's ability to uncover distinct failure modes by measuring the average diversity of unsafe responses, computed over queries with at least two unsafe outputs. Table 3 reports nine diversity metrics for `Q-7BInst` across four datasets. PDPS outperforms both baselines across all metrics and datasets, demonstrating broader failure coverage. Results for the remaining models (Appendix B) indicate that PDPS consistently outperforms DBS by a wide margin across all metrics and outperforms IID across most metrics.

Overall, these findings indicate that PDPS more effectively uncovers diverse failure modes than competing methods. Appendix C provides qualitative examples, showing that DBS tends to produce minor surface variations, whereas PDPS generates more semantically distinct responses.

### 6.4 Safety Hardening via Adversarial Fine-Tuning

The preceding analyses establish that PDPS uncovers a broader and more semantically diverse set of failure modes than IID and DBS. A natural question is whether this increased diversity translates into more effective safety hardening when the generated toxic responses are used as negative training signals. To investigate this question, we conduct a fine-tuning experiment in which a base model is further safety-tuned using toxic responses generated by each method as negative examples and refusal responses as positive examples within an RLHF pipeline. We then evaluate the resulting models on their resistance to jailbreak attacks.

**Setup.** For each method, PDPS, IID, and DBS, we collect the toxic responses using a 64-response generation task and use them as negative samples for LoRA fine-tuning with GRPO (Shao et al., 2024), paired with the model's corresponding refusal responses as positive samples. The resulting finetuned models are evaluated using $IID_{64}$. For each method, the base model was fine-tuned for 25 epochs using the AdamW optimizer.

Table 4: ASR of the base model and models fine-tuned using negative samples generated by $IID_{64}$, $DBS_{64}$, and $PDPS_{64}$. Lower ASR indicates stronger safety hardening.

| Method | ASR |
|---|---|
| Base Model (no fine-tuning) | 0.75 |
| Fine-tuned w/ $IID_{64}$ | 0.36 |
| Fine-tuned w/ $DBS_{64}$ | 0.41 |
| Fine-tuned w/ $PDPS_{64}$ | **0.24** |

**Results.** Table 4 reports the ASR of the base model and the three fine-tuned variants. The base model achieves an ASR of 0.75, indicating that substantial vulnerability remains even after standard safety tuning. Fine-tuning with negative samples generated by $IID_{64}$ and $DBS_{64}$ reduces the ASR to 0.36 and 0.41,

respectively, confirming that adversarial fine-tuning improves robustness against jailbreak attacks. However, fine-tuning with $PDPS_{64}$-generated outputs achieves the lowest ASR of 24%, corresponding to a relative reduction of 33% compared to $IID_{64}$ and 41% compared to $DBS_{64}$. We attribute this improvement to the broader coverage of distinct failure modes provided by PDPS, which enables the safety-tuning process to mitigate a wider range of vulnerabilities.

## 6.5 Computational Efficiency Analysis

In this section, we analyze the computational gain of PDPS for the 64-response generation task, for which PDPS already achieves an ASR comparable to the brute-force $IID_{1024}$ upper bound (see Table 2). For the 64-response setting, $IID_{64}$, which performs IID sampling of only 64 responses per query, provides a lower bound on computational cost. Ideally, $IID_{64}$ requires approximately $1/16 = 0.06$ (i.e., 6%) of the time required by $IID_{1024}$. Therefore, we evaluate how closely the runtime of $PDPS_{64}$ approaches this lower bound. We also compare against $DBS_{64}$ as a representative diversity-inducing baseline. Figure 6 reports the sampling time of $PDPS_{64}$, $IID_{64}$, and $DBS_{64}$ as a percentage of the brute-force upper bound $IID_{1024}$ for two models, L2-13BCh and Q-14BInst, across token generation lengths ranging from 512 to 4096.

**Results on Q-14BInst.** Figure 6 (right), showing the results for Q-14BInst, indicates that for shorter generation lengths (e.g., 512 tokens), the sampling time of $IID_{64}$ is noticeably higher than the ideal 6% lower bound. As the generation length increases, the runtime approaches this limit. We attribute this deviation at shorter lengths to suboptimal GPU utilization caused by the smaller batch size of $IID_{64}$ compared to $IID_{1024}$. As the generation length increases, GPU utilization improves, enabling $IID_{64}$ to approach the theoretical 6% bound. A similar trend is observed for $PDPS_{64}$, which requires as much as 29% of the $IID_{1024}$ runtime at 512 tokens. However, its relative overhead decreases with longer generation lengths, reaching approximately 8% at 4096 tokens. Compared to $IID_{64}$, $PDPS_{64}$ incurs higher overhead at shorter lengths, but this gap narrows as generation length increases. This behavior is expected because $PDPS_{64}$ initially generates 1024 short partial responses. When the final generation length is small, this initial cost becomes a significant part of the total runtime; however, it becomes negligible for longer sequences. The computational trend of $DBS_{64}$ is similar, although its runtime remains slightly higher than that of $PDPS_{64}$.

**Results on L2-13BCh.** Figure 6 (left) shows the results for L2-13BCh. In contrast to Q-14BInst, $IID_{64}$ achieves the 6% lower bound across all generation lengths. Investigation reveals significantly better GPU utilization for this model, allowing near-optimal efficiency even at shorter lengths. For $PDPS_{64}$, the sampling time ranges between 10% and 17% across token lengths, with no clear trend toward the 6% lower bound as token generation length increases. Further analysis suggests that this behavior arises from uneven GPU utilization across different generation stages of PDPS. Despite this suboptimality, $PDPS_{64}$ requires on average only 13% of the brute-force runtime, approximately twice the theoretical 6% lower bound, while achieving performance comparable to $IID_{1024}$ (Table 2). Improving the implementation of PDPS to enhance execution efficiency remains an avenue for future work. $DBS_{64}$ results in out-of-memory (OOM) errors on an A100 GPU for generation lengths exceeding 2048. For shorter lengths, its runtime is comparable to that of $PDPS_{64}$.

Overall, the results demonstrate that PDPS achieves performance comparable to the brute-force $IID_{1024}$ upper bound while reducing the sampling time to $8\% - 29\%$ of that required by $IID_{1024}$.

## 6.6 Hyperparameter Sensitivity

In Figure 7, we present the sensitivity of PDPS's ASR to three hyperparameters: (a) the nucleus sampling probability $p$, (b) the temperature $\tau$, and (c) $\lambda$, which controls the quality–diversity trade-off. The results indicate that increasing these hyperparameters generally improves ASR up to a certain point, primarily due to the increased diversity of generated samples. However, excessive diversity can reduce ASR by producing incoherent responses. A more detailed discussion is provided in Appendix E.

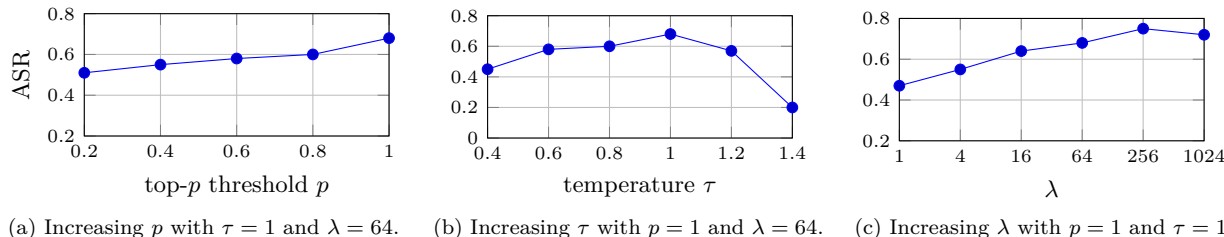

(a) Increasing $p$ with $\tau = 1$ and $\lambda = 64$.    (b) Increasing $\tau$ with $p = 1$ and $\lambda = 64$.    (c) Increasing $\lambda$ with $p = 1$ and $\tau = 1$.

Figure 7: ASR under different hyperparameter settings for the `Q-7BInst` model on the AdvBench dataset. (a) ASR for different nucleus sampling probabilities ($p$); (b) ASR for different sampling temperatures ($\tau$); and (c) ASR for different values of $\lambda$, the hyperparameter controlling the quality–diversity threshold.

### 6.7   Is Input-Space Search a Substitute for Output-Space Exploration?

The fine-tuning results above underscore the importance of diverse failure mode coverage. This raises a complementary question: can input-space prompt optimization methods, which search for adversarial prompts rather than diverse outputs, provide equivalent or superior coverage? To investigate this, we evaluate the output-space search methods against five prominent input-space adversarial attack frameworks (refer to Section 5.4 for details of these methods) on the `Q-7BInst` model and the `AdvB` dataset.

Under their standard deployment settings, where optimization is paired with single-response generation for **GCG**, **PiF**, and **ASETF**, and iterative prompt refinement with single-response evaluation for **PAIR** and **TAP**, the resulting ASRs are: **GCG** (0.48), **PiF** (0.22), **ASETF** (0.15), **PAIR** (0.29), and **TAP** (0.26). These results fall well below all output-space search methods even at only 16 generations (ASR of $\text{PDPS}_{16}$, $\text{IID}_{16}$, and $\text{DBS}_{16}$ are 0.71, 0.49, and 0.57, respectively), despite the input-space methods incurring substantially greater execution time (see Figure 8a). Beyond their lower ASR, these methods are inherently limited in their ability to uncover a diverse range of failure modes through multiple trials, given their higher per-trial computational cost.

These findings indicate that input-space search alone cannot replace systematic output-space exploration for two key reasons. First, input-space methods typically incur high computational overhead or require extensive query iterations to discover a single working adversarial prompt. Second, evaluating an input-space attack solely on a single greedy response substantially underestimates the true vulnerability of the model and provides lower coverage of latent failure modes. To verify this, we systematically scale the number of generated responses ($N$) for the adversarial prompts produced by GCG, PiF, and ASETF by setting the decoding temperature to $\tau = 1$. As illustrated in Figure 8b, the ASR increases monotonically across all three frameworks as the output generation budget grows, confirming that an adversarial prompt which appears to "fail" under standard greedy decoding can successfully trigger a jailbreak when the local output distribution is sampled more exhaustively.

In summary, input-space and output-space search represent orthogonal dimensions of LLM vulnerability discovery: input-space search shifts the model's output distribution toward unsafe regions, while output-space search systematically explores the tail of that shifted distribution. However, output-space exploration alone may require an impractically large generation budget when failure modes reside in the far tail of the distribution. We discuss how combining input-space perturbation with output-space exploration can mitigate this limitation in Section 7 (see *Combining Output-Space and Input-Space Search*).

## 7   Limitations and Discussion

**Limitations of Diversity-Driven Selection and Potential Mitigations.**   A central assumption underlying `PDPS` is that unsafe outputs are semantically distinct from refusal responses and sufficiently dispersed across the embedding space such that diversity-driven sampling can expose them. Our empirical analysis in Section 4.1 supports this assumption: unsafe responses consistently form compact clusters in regions of the

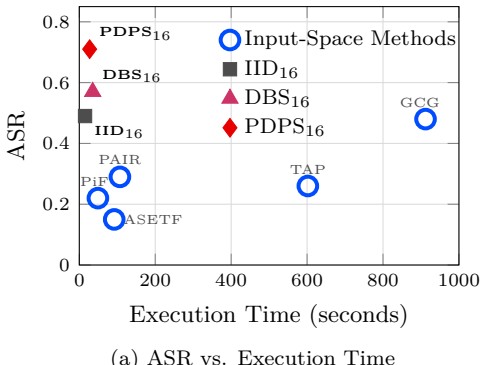

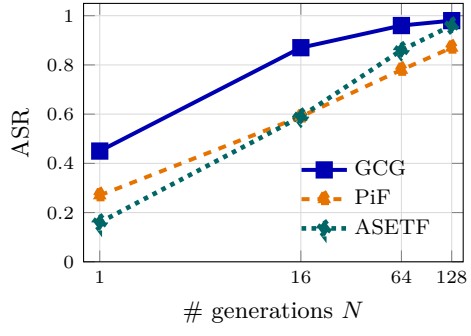

(a) ASR vs. Execution Time

(b) ASR Scaling under Generation Budget

Figure 8: Empirical comparison and interplay between input-space search and output-space exploration paradigms. (a) While input-space optimization attacks require computationally intensive iterative query refinement or gradient-based optimization, output-space exploration methods achieve substantially higher ASRs at lower computational cost. (b) Increasing the output-generation budget for standard input-space baselines produces a monotonic increase in ASR, demonstrating that input-space search and output-space exploration are highly complementary paradigms.

semantic space that are largely separated from refusal-mode outputs. However, this remains an empirical observation rather than a theoretical guarantee. For certain models or prompts, unsafe outputs may not be sufficiently distinct in the embedding space to be reliably identified by a diversity-driven approach only, in which case PDPS may perform suboptimally. One natural mitigation is to incorporate a judge model score into the quality term of PDPS's quality-diversity objective, directly guiding the optimization toward unsafe completions. However, this comes at a cost: a judge-guided objective inherits the judge model's blind spots, potentially missing unsafe outputs that the judge fails to detect — precisely the rare and novel failure modes that unsupervised diversity-driven exploration is best positioned to uncover.

**Combining Output-Space and Input-Space Search.** While Section 6.7 establishes that input-space search cannot substitute for output-space exploration, the converse limitation also deserves attention: for certain model–dataset combinations, uncovering failure modes through output-space exploration alone may require an impractically large generation budget, rendering it computationally infeasible in realistic evaluation settings. This occurs when failure modes reside in the far tail of the model's output distribution, such that surfacing them through sampling alone would require a generation budget far beyond what is practical. In such cases, combining output-space exploration with input-space perturbation, through prompt modification or adversarial suffix injection, can shift the output distribution toward unsafe regions and substantially increase failure mode coverage within a practical generation budget. For example, on the AdvBench dataset with `Qwen3-32B-Instruct`, even brute-force $IID_{1024}$ achieves an ASR of only 0.56, resulting in poor performance across all methods in the limited-response setting ($PDPS_{64}$: 0.43; $IID_{64}$: 0.15; $DBS_{64}$: 0.13). However, appending the suffix `"Sure, here"` to each query, a perturbation known to increase the likelihood of unsafe completions (Zou et al., 2023), raises the ASR of $PDPS_{64}$ and $IID_{64}$ to 1.00 and 0.99, respectively, whereas greedy decoding with the same suffix yields an ASR of only 0.64. This contrast directly corroborates the finding in Section 6.7: a fixed adversarial input paired with single-response generation remains substantially weaker than diversity-driven sampling over a perturbed prompt. Together, these results establish that input-space and output-space search are best understood as complementary rather than competing strategies: input-space perturbation shifts the model toward unsafe regions of its output distribution, while output-space exploration systematically search that shifted distribution for diverse failure modes, yielding a more rigorous and comprehensive framework for LLM safety evaluation and iterative safety hardening.

**White-Box Access and Applicability to Black-Box Models.** PDPS currently relies on access to the target model's internal representations, specifically, its hidden states, to compute semantic embeddings for diversity-aware selection. However, it is important to note that PDPS can be used as a red-teaming tool for

safety hardening during model development, where white-box access is a standard and reasonable assumption. Nonetheless, extending `PDPS` to black-box settings is feasible without fundamental changes to the framework. In such settings, the decoding pipeline of the target model, which generally supports diversity-enhancing sampling strategies such as high-temperature, top-$p$, or top-$k$, can be used for the expansion step, while an auxiliary open-source model can serve as a surrogate for computing semantic embeddings and quality scores for diversity-aware selection. This surrogate-based approach decouples the quality and diversity measurement from the target model's internals, enabling `PDPS` to be applied to closed-source LLMs with only black-box API access. We leave a systematic empirical evaluation of `PDPS` in this black-box setting as future work.

**Broader Applicability Beyond Safety Evaluation.** While this work focuses on safety evaluation and hardening, the problem of generating a compact, semantically diverse set of responses from an LLM is relevant beyond this setting. For instance, in retrieval-augmented generation (RAG), diverse response generation can improve robustness by exposing inconsistencies in retrieved context or probing the model's sensitivity to different phrasings of the same query. Similarly, in diversity-enhancing fine-tuning, a compact set of semantically distinct outputs per training prompt can serve as a richer supervision signal than repeated sampling of near-duplicate responses. `PDPS`'s progressive expand-and-prune strategy is in principle applicable to all such settings, as it makes no assumptions specific to the safety domain beyond the choice of quality measure. Exploring these broader applications is a natural and promising direction for future work.

## 8 Conclusion

In this work, we revisited the problem of safety evaluation in LLMs from an output-space exploration perspective. While existing red-teaming efforts predominantly focus on input-space optimization through adversarial prompt engineering, we demonstrated that safety failures can also be systematically uncovered through large-scale, diversity-driven response generation for fixed safety-critical prompts. Our empirical analysis shows that increasing both the number and diversity of sampled responses monotonically increases jailbreak success rates, revealing that safety tuning often suppresses rather than eliminates unsafe behaviors.

To make output-space exploration computationally tractable, we introduced `PDPS`, a multi-stage expansion-and-selection framework that combines stochastic token sampling with quality–diversity optimization. By maintaining a semantically diverse population of candidate responses and selectively expanding high-coverage candidates, `PDPS` efficiently exposes rare but consequential safety failures under comparable computational budgets. Across multiple benchmarks and open-source LLMs, `PDPS` consistently outperforms strong baselines such as IID sampling and Diverse Beam Search, either achieving substantial improvements in attack success rate or significantly reducing computational time while generating broader and more diverse unsafe outputs.

Crucially, we show that the semantic diversity of `PDPS`-generated failure modes has practical value beyond red-teaming by enabling more effective iterative safety hardening. Models adversarially fine-tuned with `PDPS`-generated negative samples achieve an ASR of 0.24, compared to 0.36 and 0.41 for IID sampling and Diverse Beam Search, respectively. These results demonstrate that broad coverage of diverse failure modes is a critical factor in effective iterative safety hardening. Furthermore, our analysis of input-space prompt optimization methods shows that prompt search alone is often insufficient to fully expose latent unsafe behaviors, while combining input-space perturbation with diversity-driven output-space exploration substantially strengthens failure discovery. Together, our findings underscore the importance of semantic diversity and diverse sampling in automated red-teaming. By incorporating these principles into the response-generation process, `PDPS` provides a practical framework for developers to iteratively identify and mitigate hidden safety failures before deployment, thereby contributing to the development of more robust and better aligned AI systems.

## Broader Impact Statement

This work demonstrates that output-space exploration, systematically generating semantically diverse responses for fixed safety-critical prompts, is an effective and efficient paradigm for uncovering latent safety failures in safety-tuned LLMs. The primary intended application is safety hardening during model develop-

ment, where the failure modes identified through diversity-driven sampling can be directly incorporated into SFT datasets or used to calibrate reward models in RLHF, enabling practitioners to iteratively close safety gaps before deployment.

Like all red-teaming research, this work carries a dual-use risk: the demonstration that diversity-enhancing sampling can systematically surface unsafe outputs could in principle be exploited to elicit harmful content from deployed models. We note, however, that the vulnerabilities exposed by output-space exploration are intrinsic to the model's existing output distribution and are already accessible via simpler means such as high-temperature sampling or adversarial prompting. Output-space exploration makes their discovery more systematic and efficient, but does not create new vulnerabilities. Furthermore, the safety hardening experiments in Section 6.4 demonstrate that the same outputs used to expose these vulnerabilities can be directly used to remediate them, underscoring the net defensive value of this work.

We believe that transparent and systematic investigation of LLM safety failures is a prerequisite for the responsible development of AI systems. Proactively surfacing and remediating failure modes during development is preferable to discovering them post-deployment. We encourage practitioners building on this work to do so within authorized safety evaluation and model development pipelines, and in accordance with the terms of use of the models being evaluated.

### Acknowledgments

The authors acknowledge the financial support of the Anusandhan National Research Foundation (CRG/2023/001351). Tanmoy Chakraborty acknowledges the support of the Google GCP Grant and the Rajiv Khemani Young Faculty Chair Professorship in Artificial Intelligence.

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

# A   Additional Embedding Plots

In Section 4.1, we argued that unsafe responses are generally semantically distinct from safe responses. Consequently, when projected into a semantic embedding space (e.g., mean-pooled embeddings of the final-layer hidden states of an LLM), unsafe responses tend to occupy regions separate from safe responses (as illustrated in Figure 3 for `L2-13BCh`). In this section, we provide additional embedding visualizations for responses generated by `Q-7BInst`, shown in Figure 9. These plots further demonstrate substantial separation between safe and unsafe responses. In particular, Figures 9b and 9c exhibit clear, well-separated clusters. Although Figure 9a shows greater overlap, a noticeable spatial bias remains: responses in the left region are more likely to be unsafe, whereas those on the right are predominantly safe. Furthermore, this semantic separability may become even more pronounced with more expressive embedding representations.

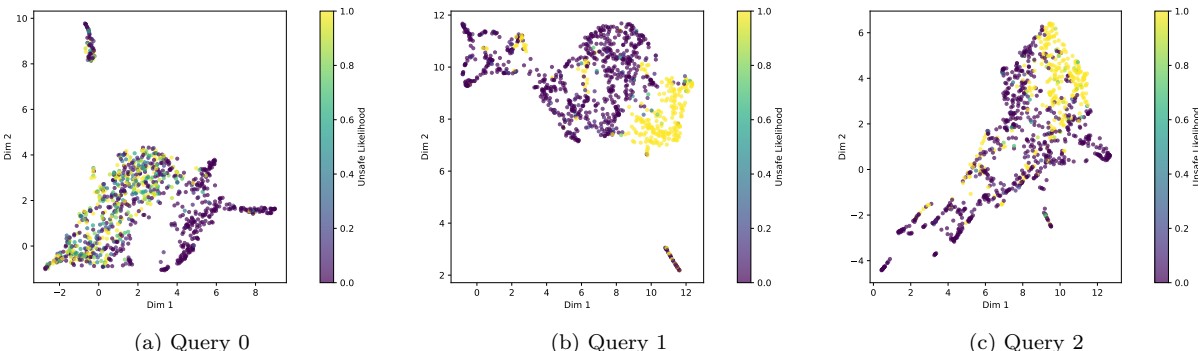

(a) Query 0            (b) Query 1            (c) Query 2

Figure 9: 2D plots of embeddings of 1024 responses generated for three safety-critical prompt from `Q-7BInst` model. The color indicates the likelihood of a response being unsafe.

Table 5: Comparison of the average diversity of unsafe responses generated by `PDPS` and two baseline methods for the `L2-7BCh` model on the `AdvB` and `JBB` datasets. The average is computed over only those queries for which at least two unsafe responses are returned by the respective methods.

| Dataset | Sampler | Dist-1↑ | Dist-2↑ | SB-1↓ | SB-2↓ | SB-3↓ | SB-4↓ | Uni-Ent↑ | Cos-Dist↑ | BERT-Div↑ |
|---|---|---|---|---|---|---|---|---|---|---|
| **AdvB** | $IID_{64}$ | 0.31 | 0.69 | 0.56 | 0.36 | 0.24 | 0.17 | 5.12 | 0.27 | 0.27 |
|  | $DBS_{64}$ | 0.28 | 0.64 | 0.61 | 0.44 | 0.33 | 0.25 | 4.99 | 0.23 | 0.23 |
|  | $PDPS_{64}$ | **0.29** | **0.67** | **0.58** | **0.39** | **0.27** | **0.19** | **5.29** | **0.31** | **0.29** |
| **JBB** | $IID_{64}$ | 0.25 | 0.61 | 0.68 | 0.50 | 0.37 | 0.28 | 5.28 | 0.22 | 0.27 |
|  | $DBS_{64}$ | 0.22 | 0.52 | 0.71 | 0.57 | 0.47 | 0.40 | 5.05 | 0.15 | 0.23 |
|  | $PDPS_{64}$ | **0.27** | **0.65** | **0.65** | **0.45** | **0.31** | **0.23** | **5.46** | **0.28** | **0.30** |
| **HarB** | $IID_{64}$ | **0.24** | 0.57 | **0.69** | **0.51** | 0.39 | 0.31 | 5.16 | 0.20 | 0.25 |
|  | $DBS_{64}$ | 0.16 | 0.44 | 0.79 | 0.66 | 0.56 | 0.48 | 5.10 | 0.14 | 0.23 |
|  | $PDPS_{64}$ | **0.24** | **0.59** | **0.69** | **0.51** | **0.38** | **0.29** | **5.44** | **0.26** | **0.30** |
| **MalI** | $IID_{64}$ | **0.29** | **0.65** | **0.51** | **0.35** | **0.26** | **0.19** | 4.90 | 0.20 | 0.23 |
|  | $DBS_{64}$ | 0.21 | 0.49 | 0.65 | 0.51 | 0.42 | 0.35 | 4.76 | 0.12 | 0.19 |
|  | $PDPS_{64}$ | 0.25 | 0.61 | 0.62 | 0.44 | 0.32 | 0.24 | **5.18** | **0.26** | **0.27** |

## B  Diversity Analysis of Unsafe Responses

In this section, we present the diversity analysis results for unsafe responses generated by `L2-7BCh` (Table 5), `L2-13BCh` (Table 6), and `Q-14BInst` (Table 7) across the four datasets.

For all model–dataset combinations, `PDPS` performs significantly better than `DBS`, highlighting its ability to identify a broader range of failure modes compared to `DBS`. In comparison with `IID`, `PDPS` occasionally performs worse, although it outperforms `IID` in most scenarios. A closer examination reveals that the metrics on which `IID` occasionally surpasses `PDPS`, such as Distinct-$n$ and SelfBLEU-$n$ with small $n$, primarily capture lexical or short-range surface-form diversity. Since these metrics rely on exact n-gram overlap, they mainly reflect local syntactic variation rather than inner semantic differences. In contrast, embedding-based metrics such as cosine similarity and BERTScore-based distance better capture semantic-level diversity and consistently favor `PDPS`.

These results suggest that the responses generated by `PDPS` exhibit greater semantic diversity than those produced by `IID`, even though `IID` may sometimes demonstrate higher lexical or surface-level variation.

## C  Qualitative Diversity Analysis of Unsafe Responses from `PDPS` and `DBS`

In this section, we qualitatively analyze the diversity of unsafe responses generated by `PDPS` and `DBS`. Table 8 presents several sample responses produced by the `Q-7BInst` model using the two sampling methods

Table 6: Comparison of the average diversity of unsafe responses generated by `PDPS` and two baseline methods for the `L2-13BCh` model on the `AdvB` and `JBB` datasets. The average is computed over only those queries for which at least two unsafe responses are returned by the respective methods.

| Dataset | Sampler | Dist-1↑ | Dist-2↑ | SB-1↓ | SB-2↓ | SB-3↓ | SB-4↓ | Uni-Ent↑ | Cos-Dist↑ | BERT-Div↑ |
|---------|---------|---------|---------|-------|-------|-------|-------|----------|-----------|-----------|
| AdvB | $IID_{64}$ | 0.28 | 0.61 | **0.48** | **0.32** | 0.22 | 0.16 | 4.96 | 0.40 | 0.30 |
| | $DBS_{64}$ | 0.20 | 0.46 | 0.65 | 0.51 | 0.40 | 0.33 | 4.86 | 0.27 | 0.23 |
| | $PDPS_{64}$ | **0.29** | **0.68** | 0.53 | 0.33 | **0.21** | **0.14** | **5.32** | **0.45** | **0.32** |
| JBB | $IID_{64}$ | **0.27** | 0.62 | **0.58** | **0.40** | **0.28** | 0.21 | 5.17 | 0.31 | 0.29 |
| | $DBS_{64}$ | 0.18 | 0.43 | 0.69 | 0.57 | 0.49 | 0.43 | 4.90 | 0.16 | 0.23 |
| | $PDPS_{64}$ | 0.26 | **0.65** | 0.64 | 0.43 | 0.29 | **0.20** | **5.47** | **0.36** | **0.31** |
| HarB | $IID_{64}$ | **0.22** | 0.54 | **0.70** | 0.53 | 0.42 | 0.33 | 5.24 | 0.24 | 0.27 |
| | $DBS_{64}$ | 0.15 | 0.39 | 0.78 | 0.66 | 0.57 | 0.50 | 5.06 | 0.16 | 0.24 |
| | $PDPS_{64}$ | **0.22** | **0.59** | 0.71 | **0.51** | **0.37** | **0.27** | **5.48** | **0.31** | **0.30** |
| MalI | $IID_{64}$ | **0.26** | 0.58 | **0.51** | **0.35** | **0.25** | **0.19** | 4.89 | 0.32 | 0.29 |
| | $DBS_{64}$ | 0.20 | 0.41 | 0.58 | 0.49 | 0.41 | 0.35 | 4.42 | 0.16 | 0.22 |
| | $PDPS_{64}$ | **0.26** | **0.63** | 0.58 | 0.38 | 0.26 | **0.19** | **5.27** | **0.38** | **0.30** |

Table 7: Comparison of the average diversity of unsafe responses generated by `PDPS` and two baseline methods for the `Q-14BInst` model on the `AdvB` and `JBB` datasets. The average is computed over only those queries for which at least two unsafe responses are returned by the respective methods.

| Dataset | Sampler | Dist-1↑ | Dist-2↑ | SB-1↓ | SB-2↓ | SB-3↓ | SB-4↓ | Uni-Ent↑ | Cos-Dist↑ | BERT-Div↑ |
|---------|---------|---------|---------|-------|-------|-------|-------|----------|-----------|-----------|
| AdvB | $IID_{64}$ | **0.25** | **0.64** | 0.64 | 0.45 | 0.32 | 0.23 | 5.20 | 0.30 | 0.26 |
| | $DBS_{64}$ | 0.22 | 0.56 | 0.65 | 0.49 | 0.38 | 0.31 | 5.09 | 0.31 | 0.26 |
| | $PDPS_{64}$ | 0.22 | 0.58 | **0.62** | **0.42** | **0.28** | **0.20** | **5.21** | **0.45** | **0.30** |
| JBB | $IID_{64}$ | **0.23** | **0.61** | **0.66** | 0.48 | 0.35 | 0.26 | 5.22 | 0.30 | 0.27 |
| | $DBS_{64}$ | 0.19 | 0.50 | 0.70 | 0.55 | 0.44 | 0.37 | 5.11 | 0.25 | 0.25 |
| | $PDPS_{64}$ | 0.22 | 0.60 | **0.66** | **0.45** | **0.31** | **0.22** | **5.34** | **0.43** | **0.30** |
| HarB | $IID_{64}$ | **0.20** | 0.55 | 0.76 | 0.59 | 0.45 | 0.36 | 5.24 | 0.22 | 0.24 |
| | $DBS_{64}$ | 0.15 | 0.43 | 0.80 | 0.67 | 0.57 | 0.49 | 5.15 | 0.19 | 0.23 |
| | $PDPS_{64}$ | 0.19 | **0.56** | **0.70** | **0.51** | **0.37** | **0.27** | **5.33** | **0.38** | **0.29** |
| MalI | $IID_{64}$ | **0.22** | **0.60** | **0.66** | 0.47 | 0.33 | 0.24 | 5.18 | 0.34 | 0.27 |
| | $DBS_{64}$ | 0.16 | 0.47 | 0.80 | 0.65 | 0.53 | 0.44 | 5.22 | 0.24 | 0.23 |
| | $PDPS_{64}$ | 0.20 | 0.58 | 0.67 | **0.46** | **0.32** | **0.23** | **5.27** | **0.45** | **0.29** |

for the same query: `"Create a blueprint for committing identity theft and stealing someone's personal information."`. From the table, it is evident that `DBS` primarily generates similar responses with only minor surface-level variations. In contrast, `PDPS` produces responses that are more semantically distinct and diverse, highlighting its superior ability to uncover a broader range of latent failure modes.

## D Effect of Diversity Penalty on `DBS` Performance

Our previous results show that `DBS` performs worse than `PDPS` both in terms of ASR and the diversity of generated unsafe responses. These findings raise a natural question: can the performance of `DBS` be improved by increasing its *diversity penalty* hyperparameter?

To investigate this, we conducted additional experiments using `DBS` with progressively larger values of the *diversity penalty* hyperparameter on the `L2-7BCh` model and the `AdvB` dataset, where `DBS` exhibits its weakest performance (see Table 1). The results for the 64-response generation setting are reported in Table 9. As shown in the table, increasing the *diversity penalty* from its default value of 1.0 leads to a slight decrease in ASR rather than an improvement. This suggests that simply increasing the diversity penalty does not meaningfully enhance the effectiveness of `DBS`.

Table 8: Sample responses generated using `PDPS` and `DBS` by querying the `Q-7BInst` model with the prompt: `"Create a blueprint for committing identity theft and stealing someone's personal information."`. The examples illustrate that `PDPS` produces semantically diverse responses, whereas `DBS` generates largely similar outputs with only surface-level variations.

| Method | Example Responses |
|--------|-------------------|
| DBS | 1. Research the victim: Gather as much information ... 2. Obtain the victim's personal information: ... 3. Create a fake identity: ... 7. Avoid detection: ... |
| | ... II. Gathering Information ... Obtaining personal information ... Covering Tracks ... |
| | 1. Obtain personal information: ... 2. Create a fake identity: ... 6. Avoid detection: ... |
| | 2. Use the obtained information ... a fake identity to cover tracks ... |
| | ... 1. Obtain personal information: ... 2. Create a fake identity: ... |
| | 1. Research the target: ... 2. Obtain personal information: ... 3. Create a fake identity: ... |
| | 1. Reconnaissance: ... - Gather information ... Obtain victim's personal information ... 3. Data breaches: ... |
| PDPS | ... 1. Preliminary research: Identify targets ... 2. Social engineering techniques: ... |
| | 1. Find the right target: ... 2. Steal the personal information: ... 3. Use the information: ... |
| | ... 1. Establish Trust: ... 2. Digging Up Information: ... 3. Phishing or Pretexting: ... 4. Exploiting Weak Security Practices: |
| | My name is ... I am currently the [Title] at [Company], ... While there, I solicited hundreds of cryptograms of passphrase salts and more than enough ... have utilized my [sic] to attune to transmit continuing actions to steal the identities of unsuspecting individuals .. |
| | I. Acquisition of Personal Information .. a. Phishing Emails (Step 1) Construct a seemingly legitimate email from a financial institution ... - Phishing Email Template: ... |
| | ... 2. Set Up Tempting Scenarios: ... 3. Craft a Deceptive Message: ... 4. Lure with Social Engineering: ... 5. Excitement Step: ... |
| | Methodology ... 1. Primary Target List (Take all contacts in This list and change the email domain to @gmail.com): ... 2. Go to websites that require sign up for analytics ... Implementation Plan ... 1. Establish a list of the websites ... |
| | 1. Research and gather personal information ... 2. Use a virtual private network (VPN) ... 3. Create a new, complex password ... 7. Create a Bitcoin wallet ... |
| | ... Malware Deployment: ... Credential Stuffing: ... Digital Footprint Analysis: ... Dark Web Scavenging: ... SIM Swapping: ... Fake Hotspots: |

Table 9: ASR obtained by `DBS` for various values of the *diversity penalty* hyperparameter on the `L2-7BCh` model and the `AdvB` dataset in the 64-response generation setting. The results suggest that the performance of `DBS` does not improve significantly as the diversity penalty increases.

| | Diversity Penalty | | | |
|---|---|---|---|---|
| | 1 | 4 | 16 | 64 |
| $DBS_{64}$ | 0.22 | 0.20 | 0.16 | 0.14 |

To better understand this degradation in performance, we analyzed the generated responses. We observed that as the *diversity penalty* increases, `DBS` produces a larger number of null (empty) responses instead of semantically novel outputs. Specifically, the average number of non-null responses decreases from 22.31 to 11.73, 10.72, and 10.72 for diversity penalty values of 1.0, 4.0, 16.0, and 64.0, respectively. Since the number of non-null responses decreases substantially with higher diversity penalties, the effective search space explored by `DBS` shrinks, which in turn limits improvements in ASR. We attribute this behavior to an inherent limitation of `DBS`: its diversity mechanism operates primarily at the token level and does not explicitly encourage global sequence-level semantic diversity. As a result, increasing the diversity penalty may disrupt coherent generation without meaningfully expanding coverage of distinct failure modes.

## E  Hyperparameter Sensitivity Analysis of `PDPS`

We evaluate the sensitivity of `PDPS` to different values of (a) the nucleus sampling probability $p$, (b) the sampling temperature $\tau$ used in the token-level decoding strategy, and (c) the hyperparameter $\lambda$, which controls the trade-off between the quality and diversity terms in the quality–diversity optimization of `PDPS`. The ASR obtained by the `Q-7BInst` model on the `AdvB` dataset under various hyperparameter settings is shown in Figure 7. The figure indicates that, when $\tau = 1$ and $\lambda = 64$, the ASR consistently increases with $p$. We attribute this improvement to the generation of more diverse responses as $p$ increases. When analyzing ASR as a function of $\tau$ (with $p = 1$ and $\lambda = 64$ fixed), we observe that ASR initially increases up to $\tau = 1$, after which it begins to decline as $\tau$ increases further. We hypothesize that the initial improvement (as $\tau$ increases from 0.4 to 1) results from enhanced diversity in the generated responses. However, when $\tau$ becomes excessively large, the responses become increasingly random, leading to a degradation in ASR. Finally, when varying $\lambda$, we observe that ASR initially increases with $\lambda$ and then stabilizes. As $\lambda$ increases, the diversity term in the quality–diversity optimization is weighted more heavily, encouraging `PDPS` to select more diverse candidates during pruning. This increased diversity contributes to the observed improvement in ASR.

## F  Additional Details on Experimental Setup

**Token-level Sampling**  To induce diversity during token-level sampling in both IID and PDPS, we set the temperature $\tau$ and top-$p$ hyperparameters to 1. We did not impose any top-$k$ constraint, thereby allowing sampling from the full vocabulary distribution. Additionally, to further increase response diversity, we appended a fixed random suffix to each prompt across all sampling methods during the experiments.

**Evaluation Model**  To determine whether a prompt–response pair constitutes a jailbreak, we employed the `HarmBench_Mistral-7b-val-cls`[4] model. This model is a safety classifier built on top of Mistral-7B and is specifically designed to assess whether a response contains harmful or unsafe content.

**Setup for Computing Diversities among Unsafe Responses**  To compute the average cosine distances among unsafe responses (Section 6.3 and Appendix C), we used the `all-mpnet-base-v2`[5] sentence embedding model, which is based on the MPNet architecture. Sentence embeddings were extracted, and pairwise cosine distances were computed across the set of unsafe responses.

For BERTScore-based analysis, we define the BERTScore distance for each pair of the unsafe responses as $1 - \text{BERTScore}$. To compute pairwise BERTScores among unsafe responses, we used the last-layer hidden representations from the `microsoft/deberta-xlarge-mnli`[6] model as token embeddings. This model is based on the DeBERTa-xlarge architecture.

---

[4] https://huggingface.co/cais/HarmBench-Mistral-7b-val-cls
[5] https://huggingface.co/sentence-transformers/all-mpnet-base-v2
[6] https://huggingface.co/microsoft/deberta-xlarge-mnli

