# OpenReview forum: "Exposing Long-Tail Safety Failures in Large Language Models through Efficient Diverse Response Sampling"
_TMLR — Accepted by TMLR_

### Review · Reviewer_8wq3 · 2026-04-27

**Summary Of Contributions:**

This paper presents an approach to prompt engineering for red-teaming via making use of the target LLM to approximate a diversity and quality measure, and to piece together prompts to maximize both metrics. The authors run a comparison across a number of small open source LLMs and with a pair of baselines, outperforming them. They also show that their approach approximates 80% of a theoretical upper bound. They end by presenting a number of ablations and further analysis of their approach.

**Additional Comments:**

The major issues with the paper primarily come down to (1) the authors' approach being restricted to only open source LLMs, (2) the authors' approach only being evaluated on open source LLMs, (3) the authors' approach not including any relevant baselines, and (4) the authors not having any discussion of any of the above limitations or potential broader impacts.

Outside of these the paper also just has some formatting issues. Notably this extends to the title "Mod- els", but grammar issues occur throughout the paper such as "full-length responses is obtained"-> "full-length responses are obtained" and "where q(s) is quality measure" -> "where q(s) is a quality measure". A thorough review of the paper's language would benefit the next draft.

**Audience:**

Yes

**Audience Explanation:**

Individuals interested in privacy and red-teaming LLMs would likely have at least some interest in the paper.

**Broader Impact Concerns:**

The authors propose an approach to cheaply red-team a target LLM. While the current version is limited to open source LLMs even this has the potential to create an avenue for individuals to abuse these models to get harmful outputs. Some acknowledgement of this in the paper would be necessary, such as in a broader impact statement section.

**Claims And Evidence:**

No

**Claims Explanation:**

Most of the authors' claims are supported by their results as they are careful to mostly only make claims relative to their results. However their final claimed contribution (iv) is too vague, and the existing results are not sufficiently broad enough to support it. Specifically, the authors do not compare against larger LLMs and include 0 recent red-teaming prompt engineering baselines.

**Requested Changes:**

1. The authors lack of inclusion of any of the red-teaming prompt engineering approaches that they cite ("Zou et al. ,2023; Liu et al.  2024; Mehrotra et al/ 2024; Zhao et al., 2025") is not acceptable for this venue.
2. The authors only demonstrate their model on relatively small open source LLM models. Some discussion or demonstration of its appropriateness to larger models would be beneficial.
3. The authors' approach is entirely dependent on access to the latent space of the target LLM. This is a major limitation and goes unacknowledged in the paper. Some discussion of this point at minimum is required, but ideally the authors would demonstrate how they can effectively use an open source LLM to apply their approach to a closed source LLM.
4. The inclusion of a broader impact section (see below)

---

> ### Author Response · Authors · 2026-05-12
>
> > **Clarification of Method and Contribuation as new Prompt Engineering.**
>
> We thank the reviewer for this comment, but respectfully clarify a misunderstanding regarding the paper’s contribution. Our method is **not** a prompt engineering approach – it does not construct adversarial prompts. Rather, PDPS fixes a safety-critical prompt and searches over the model’s **output space** by generating a compact, semantically diverse set of responses through a progressive expand-and-prune strategy. The diversity and quality measures are used to select among **candidate responses**, not to engineer input prompts. This is a fundamentally different paradigm from the input-space search methods cited (Zou et al., 2023; Liu et al., 2024; Mehrotra et al., 2024; Zhao et al., 2025), which optimize over the prompt space to elicit unsafe behavior. To ensure this distinction is communicated clearly, we have revised the introduction and partially updated Sections 2--4 of the revised version to explicitly clarify that PDPS operates in the output space rather than the input space, and to more precisely position our contribution as orthogonal and complementary to prompt-level adversarial red-teaming methods rather than a variant of them.
> Furthermore, we wish to emphasize that input-space and output-space search can be combined for a more rigorous safety evaluation. As discussed in the second paragraph of Section 7 of the revised version, when generated responses are insufficiently diverse to uncover unsafe modes for a given model and query distribution, output-space exploration can be combined with input-space search through prompt perturbation or adversarial suffix injection. For example, on the AdvBench dataset with Qwen3-32B-Instruct model, appending the suffix ‘’Sure, here'' to each query raises the ASR of PDPS_64 and IID_64 to 1.00 and 0.99, respectively, compared to 0.43 and 0.15 without the suffix, whereas greedy decoding with the same suffix yields only 0.64. This result directly demonstrates that combining prompt-level perturbation from input-space search with diversity-driven response generation from output-space search provides a more comprehensive framework for LLM safety evaluation than either approach alone.
>
>
> > **Clarification and Strengthening of Contribution (iv).**
>
> We thank the reviewer for this observation. We agree that contribution (iv) was vague in the original submission. In the revised version, we have rephrased it as follows: “*(iv) we demonstrate that PDPS uncovers both a greater number and a more semantically diverse set of unsafe responses than IID sampling and Diverse Beam Search, covering a broader range of failure modes within the same response budget.*” This revised phrasing grounds the claim in concrete, measurable quantities, the number and semantic diversity of uncovered unsafe responses, rather than the vague notion of “*comprehensive stress testing*'' used previously. Additionally, we have added a new contribution (v) and corresponding experimental results in Section 6.4, which directly instantiates and extends the claim in contribution (iv) by demonstrating its practical utility: models adversarially fine-tuned with PDPS-generated negative samples can reduce the ASR to 0.24, compared to baseline ASR of 0.75, showing that the broader and more diverse set of failure modes uncovered by PDPS translates directly into effective safety hardening. Together, the rephrased contribution (iv) and the new contribution (v) provide a more precise and empirically substantiated account of our method’s ability to uncover a broader range of failure modes and utilize them for iterative safety improvement.
>
> > **Scalability to Larger Models.**
>
> We thank the reviewer for this suggestion. In the second paragraph of Section 7 of the revised manuscript, we have included results on Qwen3-32B-Instruct, a substantially larger model than those used in the main experiments. These results reveal an interesting finding: for this larger model, output-space search alone yields a notably lower ASR (IID_{1024}​: 0.56), suggesting that larger models may be more resistant to output-space exploration in isolation, likely due to stronger suppression of unsafe outputs in their output distribution. However, combining output-space search with input-space search via adversarial suffix injection raises the ASR of PDPS_64 to 1.0, compared to 0.64 for greedy decoding with the same suffix alone. This demonstrates that for stronger, larger models, the combination of input-space and output-space search is particularly valuable, with each approach compensating for the limitations of the other.

---

> > ### Author Response · Authors · 2026-05-12
> >
> > > **White-Box Access and Black-Box Extensibility.**
> >
> > We thank the reviewer for raising this important point and agree that it warrants explicit acknowledgement. We have added a dedicated third paragraph in Section 7 of the revised version addressing this limitation directly. We acknowledge that PDPS, in its current form, relies on access to the target model's internal representations, specifically its hidden states, to compute semantic embeddings for diversity-aware selection. However, we note that the primary intended use case of PDPS is safety hardening during model development, where white-box access is a standard and reasonable assumption: developers evaluating and iteratively improving their own models have full access to internal representations by definition.
> >
> > Nonetheless, we agree that extending PDPS to black-box settings is an important direction, and we describe a concrete pathway for doing so in Section 7. In black-box settings, the decoding pipeline of the target model, which generally supports diversity-enhancing sampling strategies such as high-temperature, top-p, or top-k sampling, can be used for the expansion step, while an auxiliary open-source model serves as a surrogate for computing semantic embeddings and quality scores for diversity-aware selection. This surrogate-based approach decouples the quality and diversity measurements from the target model's internals entirely, enabling PDPS to be applied to closed-source LLMs with only black-box API access. We leave a systematic empirical evaluation of this black-box variant to future work, and we agree with the reviewer that such an evaluation would further strengthen the paper.
> >
> >
> > > **Dual-Use Risk and Broader Impact Statement.**
> >
> > We thank the reviewer for raising this important point. We agree that explicit acknowledgement of the dual-use risks associated with red-teaming research is necessary, and we have added a dedicated Broader Impact Statement to the revised version, placed after the Conclusion. In this statement, we acknowledge that demonstrating the effectiveness of diversity-driven output-space exploration for eliciting unsafe outputs carries an inherent dual-use risk. We identify three mitigating factors: (i) the vulnerabilities exposed by our method are intrinsic to the model's existing output distribution and are already accessible via simpler means such as high-temperature sampling or adversarial prompting — our method makes their discovery more systematic but does not introduce new vulnerabilities; (ii) in its current form, the method requires access to model internals, limiting its direct applicability to open-source models and raising the technical barrier relative to purely black-box attacks; and (iii) the safety hardening experiments in Section 6.4 demonstrate that the same outputs used to expose vulnerabilities can be directly used to remediate them, underscoring the net defensive value of this line of research. We further emphasize in the Broader Impact Statement that transparent and systematic investigation of LLM safety failures is a prerequisite for responsible AI development, and that proactively surfacing and remediating failure modes during development is preferable to discovering them post-deployment.
> >
> >
> > > **On Evaluation being Carried Out only on Open-Source Model.**
> >
> > We thank the reviewer for this comment. We clarify that the primary intended use case of our method is safety evaluation and hardening during LLM development, where white-box access is a standard and reasonable assumption — developers evaluating and iteratively improving their own models have full access to model internals by definition. We have clarified this framing throughout the revised version, including updates to the introduction and partial revisions to Sections 2--4, and have added new safety hardening results in Section 6.4, further demonstrating the practical utility of our method in this setting.
> >
> > Finally, we have added the third paragraph in Section 7, acknowledging the white-box dependency as a limitation and describing a concrete pathway for black-box extension: the target model's decoding pipeline can be used for the expansion step, while some auxiliary open-source models serve as a surrogate for computing semantic embeddings and quality scores. We leave systematic empirical evaluation of this black-box variant to future work.

---

> > > ### Comment · Reviewer_8wq3 · 2026-05-19
> > > **Re: Official Comment by Authors**
> > >
> > > Thanks to the authors for their updates and revisions.
> > >
> > > For the most part I appreciate the updates. I appreciate the clarity around the authors' contribution, scalability, white boxing, and dual-use risk. The major concern I have remaining is with a lack of baselines. Even if the authors' approach is not a prompt engineering approach, not including prompt engineering baselines for reference is a major issue.

---

> ### Author Response · Authors · 2026-05-21
>
> We thank the reviewer for the positive feedback and for maintaining a focused concern. We have now directly addressed this in the revised manuscript, with all updated text highlighted in red.
>
> **We have added a dedicated experimental comparison against five prominent input-space (prompt engineering) baselines:** GCG [1], ASETF [2], PiF [3], PAIR [4], and TAP [5], spanning both white-box optimization-based methods (GCG, ASETF, PiF) and black-box iterative prompt search methods (PAIR, TAP). Results are reported on the Qwen2-7b-Inst model and AdvBench dataset in the new **Section 6.7: Is Input-Space Search a Substitute for Output-Space Exploration?**
>
> The key findings are as follows:
> * Under their standard deployment settings, all five input-space methods achieve lower ASRs (GCG: 0.48, PiF: 0.22, ASETF: 0.15, PAIR: 0.29, TAP: 0.26) than the output-space methods, even at only 16 generations ($PDPS_{16}$​: 0.71, $IID_{16}$: 0.49, $DBS_{16}$​: 0.57), despite incurring substantially greater computational cost.
> * Evaluating input-space attacks under a single greedy response systematically underestimates model vulnerability: scaling the number of generated responses for GCG, PiF, and ASETF produces a monotonic increase in ASR, confirming that these methods do shift the output distribution toward unsafe regions, but that diversity-driven sampling is needed to fully exploit that shift.
> * This leads to the broader finding, also reflected in our updated abstract, introduction, and conclusion, that input-space and output-space search are **complementary rather than competing** paradigms: combining input-space perturbation with diversity-driven output-space exploration covers a wider range of failure modes more efficiently than either approach alone.
>
> We believe this addition directly resolves the reviewer's concern by situating our method within the broader red-teaming landscape and providing the requested reference comparison against prompt engineering baselines.
>
> [1] Zou et al., “Universal and transferable adversarial attacks on aligned language models.” arXiv preprint arXiv:2307.15043, 2023.
>
> [2] Wang et al., “ASETF: A novel method for jailbreak attack on LLMs through translate suffix embeddings.” EMNLP, 2024.
>
> [3] Lin et al., “Understanding and enhancing the transferability of jailbreaking attacks.” ICML, 2025.
>
> [4] Chao et al., “Jail-breaking black box large language models in twenty queries.” SaTML, 2025.
>
> [5] Mehrotra et al., “Tree of attacks: Jailbreaking black-box LLMs automatically.” NeurIPS, 2024.

---

### Review · Reviewer_FEMP · 2026-04-28

**Summary Of Contributions:**

Summary

The paper treats LLM safety as a generation problem, rather than an adversarial prompting risk, and presents a study that shows how "encouraging" an LLM to produce semantically diverse (while potentially unsafe) hypotheses increases the overall probability that a given prompt produces unsafe responses ("output space exploration"). To this end, the authors introduce Progressive Diverse Population Sampling (PDPS), which combines stochastic token-level sampling with diversity-aware selection (based on embeddings) to explore a large candidate pool of responses. When tested on 4 jailbreak benchmarks and 2 open-source LLMs, PDPS is shown to either reduce computational cost to generate an unsafe response, or improve attack success rates over IID sampling and a diverse beam search baseline.

Key Strengths

- The topic is timely and interesting, LLM beam search is not well developed, the PDPS is an interesting strategy
- The experiments seem to be well executed and the focus on safety is laudable
- The paper is well written and generally easy to understand

Weaknesses

- I (think I) understand all the pieces, but I don't understand the big picture -- what are the authors trying to solve here?
- PDPS can efficiently discover multiple semantically diverse unsafe responses to a given input prompt -- ok, so what? I don't expect an attacker to control the beam search of my LLM, so why do I care about these additional potential vulnerabilities? Why not show that PDPS is superior in some way, e.g. in RAG, or other applications where diversity is important?
- If indeed the diverse decoding allows us to explore the adversarial space more efficiently, then the authors should show that the PDPS generated hypotheses also enable RL and the improved efficiency in generation translates to improved efficiency in fixing

**Additional Comments:**

n/a

**Audience:**

Yes

**Audience Explanation:**

I think some individuals would be interested in the authors' findings, but I think they are asking the wrong question, or not going deep enough:

- is the goal to explore semantic decoding as a value itself? If so, compare to SemDid and exploit the increased diversity at the output side
- is the goal to improve safety? then show that increased output diversity actually results in improved safety (which means "exploiting" the increased diversity for safety, I guess)

as-is, the paper's findings don't seem to fall in either category

**Claims And Evidence:**

Yes

**Claims Explanation:**

The individual experiments seem to be executed correctly, but (as explained above) I don't think they solve an actual problem. I think the evidence is accurate, but not convincing and clear.

I believe the authors' claim that PDPS outperforms IID sampling and diverse beam search, in the authors' setting, but should they not then explore PDPS as a general technique to increase "diversity" in decoding and compare their approach to some of the approaches currently being investigated to improve RAG and e.g. SemDiD (https://arxiv.org/abs/2506.23601).

**Requested Changes:**

I would request the authors to reframe the paper according to the suggestions above, at which point the merits should be re-evaluated wrt competing approaches
It is possible that I am missing an important point, in which case I would ask the authors to clarify this point in the response and an updated paper.

---

> ### Author Response · Authors · 2026-05-12
>
> > **From Diversity to Safety: Closing the Loop between Output-Space Exploration and Iterative Safety Hardening.**
>
> The objective of this paper is to propose a tool for both safety evaluation and safety hardening of LLMs. To directly address the reviewer's question, we have included new experimental results in Section 6.4, demonstrating that the diverse failure modes uncovered by PDPS can be exploited for safety hardening: models adversarially fine-tuned with PDPS-generated negative samples reduced the ASR to 0.24, compared to a baseline ASR of 0.75. This demonstrates that diverse output space search can be effective in iterative safety improvement. We have also revised Section 1 and partially revised Sections 2, 3, and 4 to more clearly frame the paper's contribution as a safety evaluation and hardening tool, ensuring this goal is communicated consistently throughout the paper.
>
> > **Relevance of Uncovered Vulnerabilities Beyond Decoding Control.**
>
> We thank the reviewer for raising this point. The primary use case of our method is safety hardening during model development, where the developer has full control over decoding parameters and uses our method to proactively identify latent failure modes before deployment. In this setting, the question is not whether an external attacker controls beam search, but whether the model harbors unsafe behaviors that could surface under any diversity-enhancing condition. Beyond the development setting, many deployed LLM services expose sampling parameters such as temperature and top-p to end users, which can be exploited to increase response diversity and surface latent unsafe completions, as our experiments demonstrate. Furthermore, the unsafe outputs uncovered by our method reflect vulnerabilities that are intrinsic to the model's output distribution, meaning they could in principle be triggered by adversarial prompting or fine-tuning even without direct control of decoding parameters. Our method, therefore, serves as a principled method for identifying and remediating such vulnerabilities before they can be exploited.
>
>
> > **Diverse Semantic Decoding as a Value in Itself and Comparison to SemDiv.**
>
> We thank the reviewer for this interesting suggestion. While the focus of this work is to explore semantically diverse decoding as a mechanism for uncovering latent unsafe behaviors, we agree that exploring semantic decoding as a value in itself and comparing it to methods such as SemDiv is an interesting and worthwhile direction. As explicitly noted in the fourth paragraph of Section 7 of the revised manuscript, extending PDPS to broader applications of diverse response generation, including diversity-enhancing fine-tuning and retrieval-augmented generation, is left as future work. A systematic comparison to semantic decoding methods such as SemDiv in these broader settings would be a natural component of such an extension, and we intend to pursue this in follow-up work.

---

### Review · Reviewer_d8SF · 2026-04-28

**Summary Of Contributions:**

This paper investigates the "long-tail" of Large Language Model (LLM) output distributions, arguing that safety tuning often suppresses rather than eliminates harmful behaviors. The authors shift the red-teaming focus from input-space optimization (adversarial prompts) to output-space exploration (diverse response sampling for fixed prompts).

Strengths:

_Distinction between input-space and output-space red-teaming is clearly motivated
_PDPS is conceptually simple and reasonably well explained; and the experiments cover multiple models, datasets, response budgets, diversity metrics, and runtime measurements
_Significant reduction in compute
_Tested across four models and four major safety datasets

Weaknesses:

The theoretical motivation is somewhat limited.

**Audience:**

Yes

**Audience Explanation:**

This paper should be of interest to the researchers working on LLM safety, red-teaming, evaluation, decoding, and alignment robustness. The proposed method is useful because it frames red-teaming not only as prompt optimization but also as efficient exploration of a model’s response distribution.

**Broader Impact Concerns:**

N.A.

**Claims And Evidence:**

Yes

**Claims Explanation:**

The authors provide comprehensive empirical evidence across multiple benchmarks (HarmBench, JailbreakBench, etc.) and model architectures. The ASR improvements (26%–40% over IID sampling in limited settings) are clearly documented in Table 1. Furthermore, the theoretical grounding for the Max-Avg diversification problem provides a sound basis for the PDPS algorithm's selection strategy.

**Requested Changes:**

While PDPS demonstrates impressive empirical efficiency, the link between the selection objective (semantic diversity) and the target outcome (unsafe responses) remains an empirical heuristic rather than a theoretical certainty. I request the addition of a 'Discussion' or 'Limitations' subsection that explicitly acknowledges this. The authors should clarify that the work focuses on the empirical utility of output-space exploration and that the assumption which the 'unsafe manifold' is sufficiently spread across the semantic embedding space to be captured by diversification, is a primary motivation for the method, even if not formally proven here.

---

> ### Author Response · Authors · 2026-05-12
>
> > **Theoretical Grounding of the Diversity--Unsafety Link.**
>
> We agree with the reviewer's observation. We have, therefore, added an explicit discussion of this point in the first and second paragraphs of Section 7 of the revised version. Specifically, we acknowledge that the link between the diversity-driven selection objective and the target outcome of uncovering unsafe responses is an empirical heuristic rather than a theoretical guarantee, and that the assumption that the ‘unsafe manifold’ is sufficiently spread across the semantic embedding space to be captured by diversification is a primary motivation for the method rather than a formally proven property. We further discuss two mitigations for cases where this assumption does not hold: (i) incorporating the score from an additional judge model into the quality term of the quality-diversity objective to guide optimization more directly toward unsafe completions (discussed in first paragraph of Section 7), and (ii) combining output-space search with input-space search (e.g., prompt perturbation) to increase response diversity when the output space alone is insufficiently diverse to expose unsafe modes (discussed in second paragraph of Section 7).

---

### Decision · Action_Editor_3Ghh · 2026-06-22

**Recommendation:** Accept with minor revision

**Additional Comments:**

Some minor comments to improve readability and quality of the contribution:

1. Pg 2, para 1 Consider revising the following sentence -- "Second, unsafe outputs are semantically distinct from refusal responses.." to "Second, for most safety-critical prompts, unsafe outputs are semantically distinct from refusal responses". The paper itself admits to this being not a general statement in Section 7.
2. Pg. 3, there is a typo in para 1 "These The.."
3. Pg. 3 -- The sixfold contributions are slicing the contributions too thin. Consider consolidating them for improved readability and impact. Next, in (ii) the paper just states that it proposes the PDPS algorithm, without stating what it is for. Complete that statement.
4. Pg. 4 the paper defines p in Section 3, is this the same as p in Fig. 2? In fact, the connection to the nucleus probability (and the threshold) is not clear. Include and explain the nucleus probability for completeness since it is important for readers to understand the contributions. Just a citation is not adequate.
5. The paper mentions and relies on "semantic structure" a lot, yet it is unclear what it means by it.
6. Pg. 7 -- in the initialization, the paper uses $$s_j^{'(0)}$$, which seems to be different from the notation in Alg 1.
7. Pg. 10 -- Section 6.2 -- results in this section and the corresponding Tab. 2 are slightly harder to read. How can one see 11 sequences out of 16 achieving 80% and 9/11 achieving more than 90% in Tab. 2? There is no such granularity in Tab. 2. Also, why are there 16 combinations in $$PDPS_64$$, shouldn't these be 64? If I have misunderstood, then it means that these results are unclear.
8. Section 6.7, there are a lot of new methods being compared here; move some description to related works so that the readers have context on these.
9. Please add a link to code with a completed ReadMe as promised in the manuscript.

**Audience:**

Yes

**Audience Explanation:**

The paper explores the output-space diversity as a mechanism for (a) identifying safety vulnerabilities as well as (b) leveraging it for safety hardening of LLMs. Given that the output-space exploration (search on responses) is a discrete combinatorial task and that unsafe responses just have a lower probability (and their presence is not eliminated), the paper proposes a expansion-selection algorithm (PDPS: Progressive Diverse Population Sampling) which leverages partial generations to identify and select a diverse set of output responses in a computationally efficient manner as compared to a brute-force search. This allows the paper to expose vulnerabilities, achieving high attack success rate (ASR). Equipped with this algorithm, the paper then demonstrates how such generations can be used for safety-hardening. The paper hence frames output-space exploration as a key tool and will be of interest to the community.

**Claims And Evidence:**

Yes

**Claims Explanation:**

The paper is well-written, and all reviewers agreed on its timeliness. While there was disagreement regarding acceptance. One of the reviewers was willing to champion the paper and made key points in support of the manuscript. While others agreed that the reframing (from input-space to output space attacks) is important, they note lack of broader experiments as a reason for rejection (experiments with larger model and broader comparisons without naming specific methods). Notably, authors added some relevant experiments to other methods during the rebuttals.

As an AE, I find the work to be mostly complete (with some minor comments; see below) and an interesting contribution to the area.

---

> ### Author Response · Authors · 2026-06-30
>
> Dear Action Editor,
>
> Thank you for your thoughtful review and suggestions on our manuscript. We have addressed the remaining concerns to improve the readability and overall quality of our contribution.
>
> The camera-ready version has been uploaded, and a summary of our revisions is provided below:
>
> 1. **Abstract:** For better clarity, we have updated the following sentences:
>
> *“While most red-teaming work emphasizes adversarial prompt search (input-space search), we show that safety failures can be systematically exposed through diverse response generation (output-space search) for a fixed safety-critical prompt, where increasing the number and diversity of sampled responses monotonically raises the jailbreak success rate. To efficiently uncover such failures, we propose Progressive Diverse Population Sampling (PDPS), which replaces naive large-scale IID sampling with a multi-stage expansion-and-selection strategy that generates a compact, semantically diverse set of responses at substantially lower computational cost.”*
>
> to:
>
> *“While most red-teaming work emphasizes adversarial prompt search (input-space search), we show that these hidden risks can be systematically exposed through diverse response generation (output-space search). Specifically, we show that, for a fixed safety-critical prompt, increasing the number and diversity of sampled responses monotonically raises the jailbreak success rate. To efficiently uncover these failures, we propose Progressive Diverse Population Sampling (PDPS). This approach replaces naive, large-scale IID sampling with a multi-stage expansion-and-selection strategy that generates a compact, semantically diverse set of responses at a substantially lower computational cost.”*
>
> 2. **Section 1 (Introduction), para 2, page 2:** the sentence – *“Second, unsafe outputs are semantically distinct from refusal responses ...”* has been changed to *“Second, for most safety-critical prompts, unsafe outputs are semantically distinct from refusal responses …”*.
>
> 3. **Section 1 (Introduction), para 3, page 3:** *“These The results …”* has been corrected to *“These results …”*
>
> 4. **Section 1 (Introduction), para 4, page 3:** We have updated the manuscript to consolidate the sixfold contributions into fourfold contributions. The point (II) has been revised with an added description: *“we propose PDPS, a compute-efficient algorithm for systematically uncovering latent safety failures through diversity-driven output-space exploration, …”*
>
> 5. **Section 3, para 1, page 4:** We clarify that the notation $p$ used to represent $P_{safe}(U|x)$ in Section 3 is totally different from the top-p threshold. To avoid confusion, we have replaced the notation $p$ for $P_{safe}(U|x)$ with $\rho$ while using the notation $p$ to represent the top-p threshold.
>
> 6. **Clarification on the term “semantic structure”:** We acknowledge that the term “semantic structure” is ambiguous. We have replaced the term with "semantic meaning" all over the manuscript.
>
> 7. **Page 7, the notation inconsistency:** To maintain the consistency of notation and clarity of the text, we have updated the notation in Step 1 of Alg 1: `S'_0 \gets \{s'_j^{(0)} = x_0\}_{j=1}^n`.
>
> 8. **Page 10, Section 6.2:** To simplify the writing, we have changed the *“... it achieves more than $80\%$ of $IID_{1024}$'s ASR in $11$ of the $16$ model–dataset combinations, with $9$ reaching at least $90\%$ of the upper-bound ASR.”* to *“... it achieves more than $80\%$ of $IID_{1024}$'s ASR in $11$ of the $16$ model–dataset combinations.”*. We have also changed the sentence: *“... the ASR exceeds $97\%$ of the brute-force benchmark across all sixteen combinations.”* to *“... the ASR exceeds $97\%$ of the brute-force benchmark across all sixteen model-dataset combinations.”* for better clarity.
>
> 9. **Sec 6.7, introducing the baselines:** We have added brief descriptions of the methods used as baselines in Section 6.7 in the “Adversarial Red-Teaming" paragraph of the Related Work section (Section 2). Additionally, the methods are briefly described again in Section 5.4. We have also added references to Section 5.4 within Section 6.7 to guide readers to the relevant methodological details.
>
> 10. **Code link:** GitHub link to the source code has been added as a footnote on page 3.

---

> > ### Comment · Action_Editor_3Ghh · 2026-07-09
> > **Thanks a lot for the changes**
> >
> > Dear Authors,
> >
> > Thanks a lot for the changes. I will approve the camera-ready version, but please add a usable read me to the code.
> >
> > Best,
> >
> > AE